RESEARCH COMMUNICATION

# Regulating G protein-coupled receptors by topological inversion

**Bray Denard[1], Sungwon Han[1], JungYeon Kim[1], Elliott M Ross[2], Jin Ye[1]***

[1]Department of Molecular Genetics, University of Texas Southwestern Medical Center, Dallas, United States; [2]Department of Pharmacology, Green Center for Systems Biology, University of Texas Southwestern Medical Center, Dallas, United States

**Abstract** G protein-coupled receptors (GPCRs) are a family of proteins containing seven transmembrane helices, with the N- and C-terminus of the protein located at the extracellular space and cytosol, respectively. Here, we report that ceramide or related sphingolipids might invert the topology of many GPCRs that contain a GXXXN motif in their first transmembrane helix. The functional significance of this topological regulation is illustrated by the CCR5 chemokine receptor. In the absence of lipopolysaccharide (LPS), CCR5 adopts a topology consistent with that of GPCR, allowing mouse peritoneal macrophages to migrate toward its ligand CCL5. LPS stimulation results in increased production of dihydroceramide, which inverts the topology of CCR5, preventing macrophages from migrating toward CCL5. These results suggest that GPCRs may not always adopt the same topology and can be regulated through topological inversion.
**Editorial note:** This article has been through an editorial process in which the authors decide how to respond to the issues raised during peer review. The Reviewing Editor's assessment is that major issues remain unresolved (see decision letter).
DOI: https://doi.org/10.7554/eLife.40234.001

*For correspondence:
jin.ye@utsouthwestern.edu

**Competing interests:** The authors declare that no competing interests exist.

## Introduction

G protein-coupled receptors (GPCRs) are a family of proteins containing seven transmembrane helices that are crucial for cell signaling (*Rosenbaum et al., 2009*). A common feature of GPCRs is that their N-terminus is extracellular, which means that during synthesis their N-terminal end is inserted into lumen of the endoplasmic reticulum (ER) (*Rosenbaum et al., 2009*). However, the vast majority of GPCRs do not contain an N-terminal cleavable signal peptide (*Guan et al., 1992*), the well-characterized mechanism directing N-terminus of most secretory and transmembrane proteins into ER lumen (*Zimmermann et al., 2011*). The difference in charge between the cytosolic and luminal loops surrounding the first transmembrane helix as well as the hydrophobicity of the first transmembrane helix of some GPCRs was reported to be crucial for translocation of their N-terminus into ER lumen (*Harley and Tipper, 1996*; *Wahlberg and Spiess, 1997*). However, these mechanisms are unlikely to be the only ones allowing GPCRs to be inserted into membranes with such an orientation.

We have recently analyzed the topology of a transmembrane protein called TM4SF20 (transmembrane 4 L6 family member 20). Similar to GPCRs, the N-terminus of TM4SF20 is inserted into ER lumen in the absence of a functional signal peptide (*Chen et al., 2016*). We revealed that a GXXXN motif in the first transmembrane helix of TM4SF20 was crucial to adopt such a topology, as mutating either the glycine or asparagine residue within the motif reversed the topology of the protein by exposing the N-terminus to cytosol (*Chen et al., 2016*). Remarkably, the topology of TM4SF20 is inverted by ceramide or related sphingolipids as the lipid alters the direction through which the first transmembrane helix is translocated across membranes (*Chen et al., 2016*). Since this regulatory mechanism does not flip transmembrane proteins that have already been synthesized but inverts the

**Table 1.** GXXXN motif presented in the first transmembrane helix of selected GPCRs.
The GXXXN motif present in the first transmembrane helix of human TM4SF20, chemokine receptors and MAS1 is highlighted. The amino acid numbers of the aligned residues are indicated.

| | **G**XXX**N** |
|---|---|
| TM4SF20(14–34) | SLLVLLLL**G**VVL**N**AIPLIVSL |
| CCR1(40–60) | LYSLVFVI**G**LVG**N**ILVVLVLV |
| CCR2(48–68) | LYSLVFIF**G**FVG**N**MLVVLILI |
| CCR3(58–78) | LYSLVFTV**G**LLG**N**VVVVMILI |
| CCR4(45–65) | LYSLVFVF**G**LLG**N**SVVVLVLF |
| CCR5(36–56) | LYSLVFIF**G**FVG**N**MLVILILI |
| CCR6(52–72) | AYSLICVF**G**LLG**N**ILVVITFA |
| CCR7(64–84) | MYSIICFV**G**LLG**N**GLVVLTYI |
| CCR9(54–74) | LYWLVFIV**G**ALG**N**SLVILVYW |
| CCR10(47–67) | VSLTVAAL**G**LAG**N**GLVLATHL |
| CXCR3(59–79) | LYSLLFLL**G**LLG**N**GAVAAVLL |
| CXCR4(48–68) | IYSIIFLT**G**IVG**N**GLVILVMG |
| CXCR5(57–77) | AYSLIFLL**G**VIG**N**VLVLVILE |
| CXCR6(37–57) | MYLVVFVC**G**LVG**N**SLVLVISI |
| MAS1(38–58) | VIMSISPV**G**FVE**N**GILLWFLC |

DOI: https://doi.org/10.7554/eLife.40234.004
The following source data is available for Table 1:

**Source data 1.** Proteins containing a GXXXN motif in their first transmembrnae helix.
DOI: https://doi.org/10.7554/eLife.40234.005

topology of newly synthesized proteins by changing the direction through which transmembrane helices are translocated across membranes, we designated this process as Regulated Alternative Translocation (RAT) (*Chen et al., 2016*). This process depends on the GXXXN motif, as destruction of the motif locked the protein into the inverted topology regardless of ceramide treatments (*Chen et al., 2016*).

In the current study, we show that the GXXXN motif is also present in many GPCRs. We further demonstrate that topology of CCR5 (CC chemokine receptor type 5), one of the GPCRs containing this motif, is inverted through RAT in lipopolysaccharides (LPS)-stimulated macrophages. These results suggest that functions of GPCRs may be regulated by topological inversion through RAT.

## Results

To search for other proteins that may undergo RAT, we performed a bioinformatics analysis to identify proteins that contain a GXXXN motif in their first transmembrane helix. This analysis revealed that ~100 transmembrane proteins, most of which are GPCRs, met our searching criteria (*Table 1— source data 1*). Interestingly, this list contains 13 of the 16 known CC and CXC chemokine receptors (*Table 1*), which are GPCRs that direct migration of leukocytes and lymphocytes toward their chemokine ligands (*Horuk, 2001*). We selected CCR5 for characterization, as its physiological functions as a chemokine receptor and pathological function as a co-receptor for human immunodeficiency virus (HIV) have been well characterized (*Lederman et al., 2006*). To determine whether ceramide induces the predicted topological inversion (*Figure 1A*), we fused a SNAP-tag, which can be covalently attached to a benzylguanine-derived fluorophore (*Keppler et al., 2003*), at the C-terminus of CCR5. Under normal circumstances, the C-terminus of CCR5 is in the cytosol (designated as orientation CCR5(A)) (*Rosenbaum et al., 2009*), making the SNAP-tag inaccessible to a cell-impermeable labeling reagent. If ceramide triggers RAT of CCR5 and the protein with the inverted topology still reaches plasma membranes, then the C-terminus of CCR5 with an inverted topology (designated as

CCR5(B)) is extracellular so that the SNAP-tag should be labeled by the cell-impermeable reagent. To control for variable protein expression, we normalized the fluorescent signal generated from the cell-impermeable reagent against that generated from a cell-permeable reagent, which labels the SNAP-tag regardless of its localization. Treatment with $C_6$-ceramide, a cell permeable analogue of ceramide that is metabolically converted to natural ceramide inside cells (*Denard et al., 2012*), increased this normalized value by ~70 fold in cells expressing SNAP-tag-fused CCR5 (*Figure 1B*). $C_6$-ceramide also increased this normalized value by 10- to 150-fold in cells expressing SNAP-tag-fused CCR1, CCR4, CCR10 and MAS1, all of which contain the GXXXN motif (*Table 1*). In contrast, $C_6$-ceramide did not enhance this value in cells expressing the SNAP-tag-fused β2- adrenergic receptor (β2AR), which does not contain this motif (*Figure 1B*). These results are consistent with the hypothesis that ceramide shifts the C-terminus of those GPCRs that contain the GXXXN motif from the cytosol to extracellular space. They also demonstrate that at least some of these GPCRs with the inverted topology reach the cell surface.

To further characterize the topological inversion of CCR5, we transfected cells with a plasmid encoding C-terminally Myc-tagged CCR5 and performed immunofluorescent microscopy with anti-Myc in the absence of saponin-mediated cell permeabilization followed by immunofluorescent

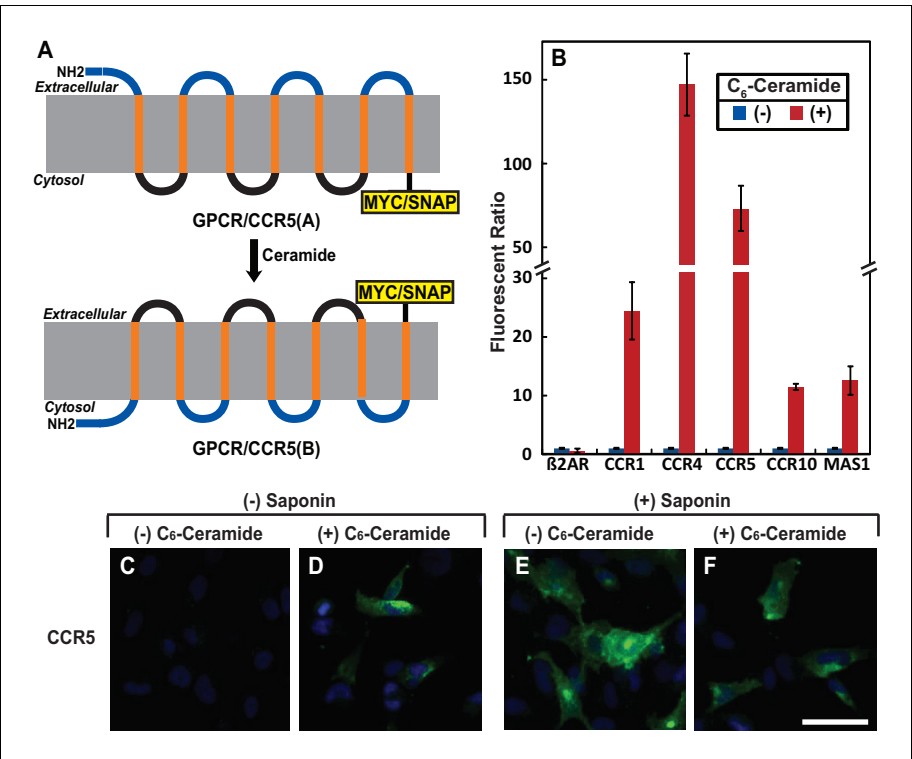

**Figure 1.** Ceramide alters localization of C-terminus of GPCRs containing the GXXXN motif. (**A**) Schematic illustration of topological inversion of C-terminally tagged GPCRs. (**B**) HEK-293 cells transfected with a plasmid encoding the indicated GPCR fused with a SNAP-tag at the C-terminus were treated with 8 μM $C_6$-ceramide for 16 hr, and labeled with a cell permeable or impermeable SNAP-tag substrate. The ratio of fluorescent signal generated from cell impermeable versus that from cell permeable substrate was reported, with the value from cells untreated with $C_6$-ceramide set at 1. Results were reported as Mean ±S.E. from triplicate incubations of a typical experiment. Similar results were obtained from two other independent experiments. (**C–F**) SV589 cells transfected with pCCR5-Myc were treated with $C_6$-ceramide as described in B, and subjected to immunofluorescent microscopy analysis with anti-Myc in the absence (**C, D**) or presence (**E, F**) of saponin-mediated cell permeabilization. Scale bar = 50 μm.

DOI: https://doi.org/10.7554/eLife.40234.002

The following figure supplement is available for figure 1:

**Figure supplement 1.** Ceramide does not increase cell permeability.
DOI: https://doi.org/10.7554/eLife.40234.003

microscopy analysis. This condition can detect CCR5 only if the C-terminal tag is extracellular but not if it is within the cytosol. The Myc-tag was undetectable in cells in the absence of ceramide, but was visible on cell surface in those treated with the lipid (*Figure 1C and D*). As a control, we performed the same analysis in cells permeabilized by saponin. CCR5 was readily detectable primarily on cell surface regardless of the ceramide treatment under this condition (*Figure 1E and F*). In the absence of saponin, ceramide did not increase the immunofluorescent signal by inducing cell permeabilization, as treatment with the lipid under this condition did not enable detection of Giantin, a Golgi marker (*Linstedt and Hauri, 1993*) (Figure 1—figure supplement 1). These results are consistent with the hypothesis that ceramide inverts the topology of CCR5.

We then determined whether ceramide altered localization of the N-terminus of CCR5. In CCR5 produced under normal circumstance (CCR5(A)), this extracellular region (*Duma et al., 2007*) contains four O-linked glycosylation sites (S6, S7, T16, S17), which are the only sites where CCR5 is glycosylated (*Bannert et al., 2001*). If ceramide induces RAT of CCR5, these sites should no longer be glycosylated as the N-terminal domain of CCR5 with the inverted topology (CCR5(B)) should be in cytosol (*Figure 2A*). Since the extracellular regions of CCR5(B) do not contain any consensus sites for glycosylation, the apparent molecular weight of CCR5(B) is expected to be lower than that of CCR5(A), because CCR5(B) should not be glycosylated (*Figure 2A*). Immunoblot analysis revealed that in the absence of ceramide, CCR5 migrated at the established molecular weight (*Figure 2B*, lane 1). Treatment with $C_6$-ceramide gradually increased the amount of another form of the protein with a lower molecular weight (*Figure 2B*, lanes 2–6).

If the lower molecular weight of CCR5(B) is caused by the lack of glycosylation, then blocking glycosylation of CCR5(A) by mutating the four O-linked glycosylation sites should abolish the reduction in molecular weight observed above even if ceramide still induces RAT of CCR5. To test this hypothesis, we mutated all four glycosylation sites to alanine (S6A, S7A, T16A, and S17A). These mutations did not block ceramide-induced RAT of CCR5 judging by the localization of C-terminus of the protein using the assay shown in *Figure 1* (*Figure 2—figure supplement 1*). The apparent molecular weight of the B form of the mutant CCR5 produced in ceramide-treated cells was the same as that of the A form produced in untreated cells (*Figure 2C*, lanes 3 and 4), and it was identical to that of wildtype CCR5(B), which was also unglycosylated (*Figure 2C*, lanes 2–4).

We also used N-terminally SNAP-tagged CCR5 known to be active in ligand binding (*Orlandi et al., 2016*) to determine the localization of the N-terminus of the protein. Consistent with the model shown in *Figure 2A*, fluorescent labeling by a cell-impermeable reagent was detected in cells incubated in the absence of $C_6$-ceramide but not those in the presence of the lipid (*Figure 2D*). Immunoblot analysis indicated that $C_6$-ceramide treatment did not affect expression of the fusion protein (*Figure 2—figure supplement 2*). It should be pointed out that the N-terminally SNAP-tagged CCR5 used in these experiments contains a signal peptide derived from CD8 at the N-terminus. Our previous study demonstrates that addition of an N-terminal signal peptide does not affect ceramide-induced RAT of TM4SF20, a transmembrane protein containing the GXXXN motif in the first transmembrane helix (*Chen et al., 2016*).

We then used an approach of cell surface cysteine labeling to determine whether ceramide inverts topology of CCR5. For this purpose, we generated a CCR5 topology reporter by mutating all five cysteine residues located at the intracellular side of CCR5(A) but leaving those located at the extracellular side intact. The reporter protein expressed in cells cultured in the absence of ceramide should be biotinylated by a cell surface sulfhydryl reactive biotinylation reagent thereby precipitable by streptavidin beads, as CCR5(A) of the reporter protein contains extracellular cysteine residues (*Figure 2E*). If ceramide inverts the topology of CCR5, then the reporter protein expressed in cells treated with ceramide is not expected to be biotinylated by the same reagent, as extracellular loops of the reporter CCR5(B) do not contain any cysteine residues (*Figure 2E*). To test this hypothesis, we incubated cells expressing the reporter CCR5 with a cell-impermeable sulfhydryl reactive biotinylation reagent, precipitated the biotinylated proteins by streptavidin-conjugated beads, and determined the amount of the reporter protein precipitated through immunoblot analysis. The reporter protein was precipitated by streptavidin-conjugated beads in the absence of $C_6$-ceramide, an observation indicating that the mutations made in the reporter protein did not affect transport of the protein to cell surface (*Figure 2F*, lanes 3 and 5). In contrast, the reporter protein was not precipitated at all in cells treated with $C_6$-ceramide (*Figure 2F*, lanes 4 and 6). As a control, we also performed the same experiment with wildtype CCR5, which contains cysteine residues on both sides of

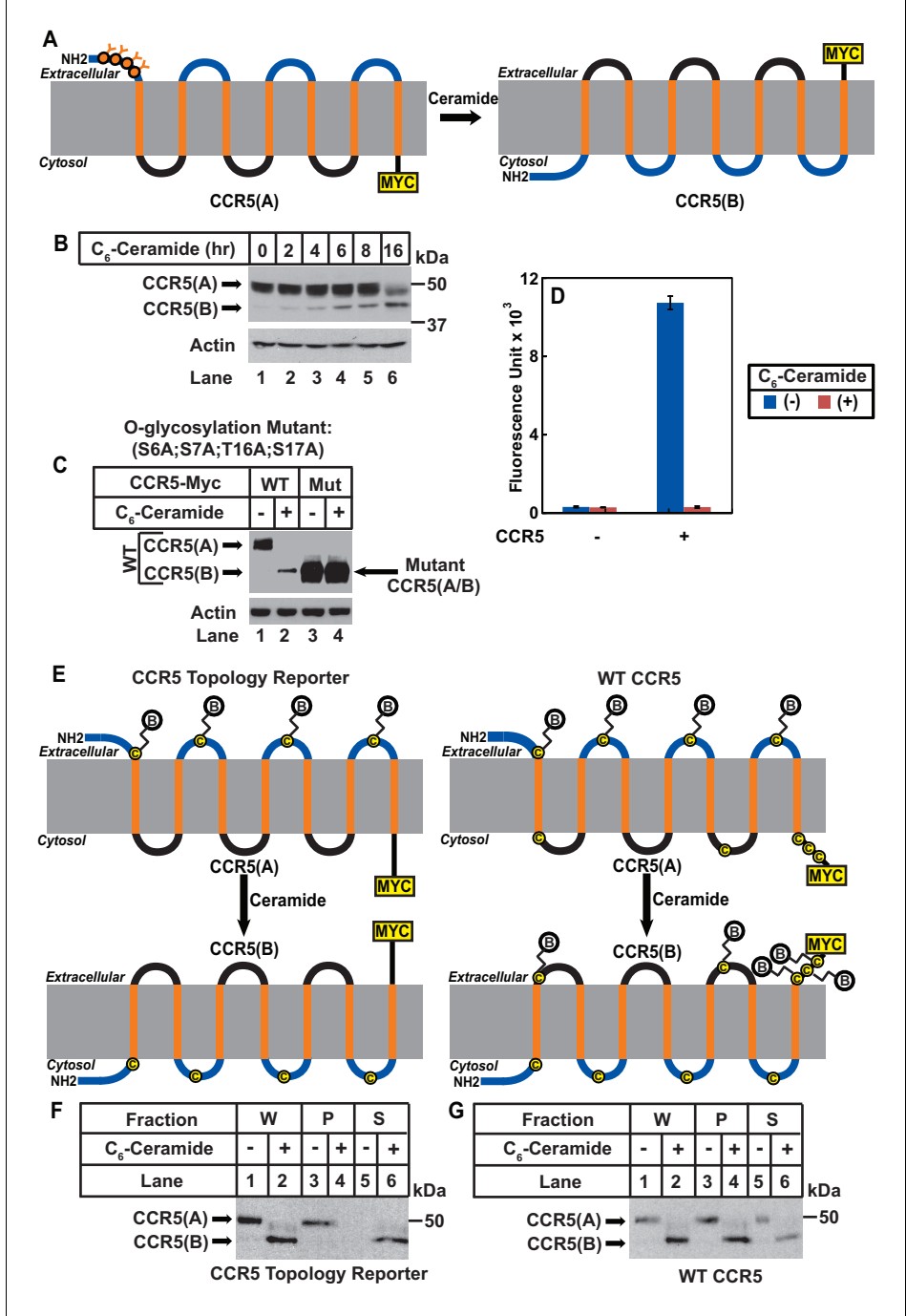

**Figure 2.** Ceramide stimulates RAT of CCR5. (**A**) Schematic illustration of the effect of RAT on O-linked glycosylation of CCR5. (**B**) SV589 cells transfected with pCCR5-myc were treated with 8 μM $C_6$-ceramide for the indicated time followed by immunoblot analysis with anti-Myc. (**C**) SV589 cells transfected with the indicated plasmid were treated with or without 8 μM $C_6$-ceramide for 16 hr followed by immunoblot analysis with anti-myc. (**D**) HEK293 cells transfected with pSNAP-CCR5 were treated with 8 μM $C_6$-ceramide for 16 hr, labeled with a cell-impermeable fluorescent substrate for SNAP-tag, and quantified the labeling reaction through a fluorimeter. Results are reported as mean ±S.E. from triplicate incubations of a typical experiment. (**E**) Schematic illustration of the effect of RAT on cell surface labeling of extracellular cysteine residues in wildtype and the topology reporter CCR5. (**F and G**) SV589 cells transfected with Myc-tagged wildtype or topology reporter CCR5 were treated with 8 μM $C_6$-ceramide for 16 hr. After cell surface labeling of extracellular cysteine residues with biotin, cell lysates were

*Figure 2 continued on next page*

*Figure 2 continued*

precipitated with streptavidin beads. Equal fractions of whole cell lysate (**W**), pellet (**P**) and supernatant (**S**) were subjected to immunoblot analysis with anti-Myc.

DOI: https://doi.org/10.7554/eLife.40234.006

The following figure supplements are available for figure 2:

**Figure supplement 1.** Mutations bocking glycosylation do not affect RAT of CCR5.

DOI: https://doi.org/10.7554/eLife.40234.007

**Figure supplement 2.** Ceramide treatment does not affect expression of SNAP-CCR5.

DOI: https://doi.org/10.7554/eLife.40234.008

membranes (*Figure 2G*). Wildtype CCR5 was precipitated by streptavidin beads regardless of ceramide treatment (*Figure 2G*). These observations demonstrated that CCR5(B) produced in the presence of ceramide was also on the cell surface, ruling out the possibility that the result shown in *Figure 2F* was caused by ceramide-induced internalization of CCR5.

We then investigated the physiological function of RAT of CCR5. LPS was reported to stimulate production of ceramide in mouse macrophage-like RAW264.7 cells (*Sims et al., 2010*). Unlike RAW264.7 cells, we observed that treatment of mouse peritoneal macrophages with LPS increased production of dihydroceramide but not ceramide (*Figure 3A*). Since dihydroceramide is structurally similar to ceramide and has been shown to perform functions previously attributed to ceramide (*Siddique et al., 2015*), we hypothesized that LPS-induced production of dihydroceramide may also induce RAT of CCR5.

To test this hypothesis, we used an antibody reacting against the second extracellular loop of CCR5(A) (*Lee et al., 1999*). This antibody recognizes a conformational epitope as it failed to detect CCR5 by immunoblot analysis but was active in identifying the protein through immunofluorescent microscopy. The immunofluorescent signal was specific to CCR5 as macrophages obtained from $Ccr5^{-/-}$ mice showed no such signal (*Figure 3—figure supplement 1*). We stained mouse peritoneal macrophages with this antibody in the absence of saponin-mediated cell permeabilization followed by immunofluorescent microscopy analysis. This condition can detect CCR5 only if the epitope was extracellular but not if it was within the cytosol. CCR5 was detected through this method in macrophages cultured in the absence but not in the presence of LPS (*Figure 3B and C*). As a control, we also performed the same analysis in permeabilized macrophages. CCR5 was readily detectable on cell surface regardless of the LPS treatment under this condition (*Figure 3D and E*, *Figure 3—figure supplement 2* for images with properly adjusted intensity).

To confirm that LPS stimulates RAT of CCR5 through increased production of dihydroceramide, we treated macrophages with fumonisin B1 (FB1), an inhibitor of ceramide synthase that catalyzes formation of the sphingolipid (*Kitatani et al., 2008*). Treatment with FB1 prevented LPS from inducing production of dihydroceramide (*Figure 3A*), and rendered the N-terminus of CCR5 to be detected by fluorescent microscopy regardless of treatment with LPS and cell permeabilization (*Figure 3F–I*).

To further determine whether LPS inverts the topology of CCR5, we performed an immunogold electron microscopy (EM) analysis to determine the localization of the epitope localized at the second extracellular loop of CCR5(A). Gold clusters presumably generated by binding of multiple gold-conjugated secondary antibodies to anti-CCR5 are considered as CCR5-specific (*Singer et al., 2001*) as such signal was never observed in CCR5$^{-/-}$ macrophages. Some CCR5 was labeled by a single gold particle, as their number in wild type macrophages was higher than that in CCR5$^{-/-}$ macrophages. However, the specificity of such labeling was difficult to determine, as these particles did exist in CCR5$^{-/-}$ macrophages. For this reason, we only analyzed CCR5 labeled by gold clusters, the number of which should be smaller than that of CCR5 molecules. In the absence of LPS, the gold particle clusterswere found on extracellular side of plasma membranes (*Figure 3J*, *Figure 3—figure supplement 3*). These particles were found on intracellular side of plasma membranes but not endocytic vesicles in cells treated with LPS (*Figure 3K*, *Figure 3—figure supplement 3*). After counting ~20 macrophages cultured in either condition, we found that the vast majority of CCR5 labeled by gold clusters had its N-terminus at the extracellular and intracellular side of plasma membranes, respectively, in cells cultured in the absence and presence of LPS (*Figure 3L*).

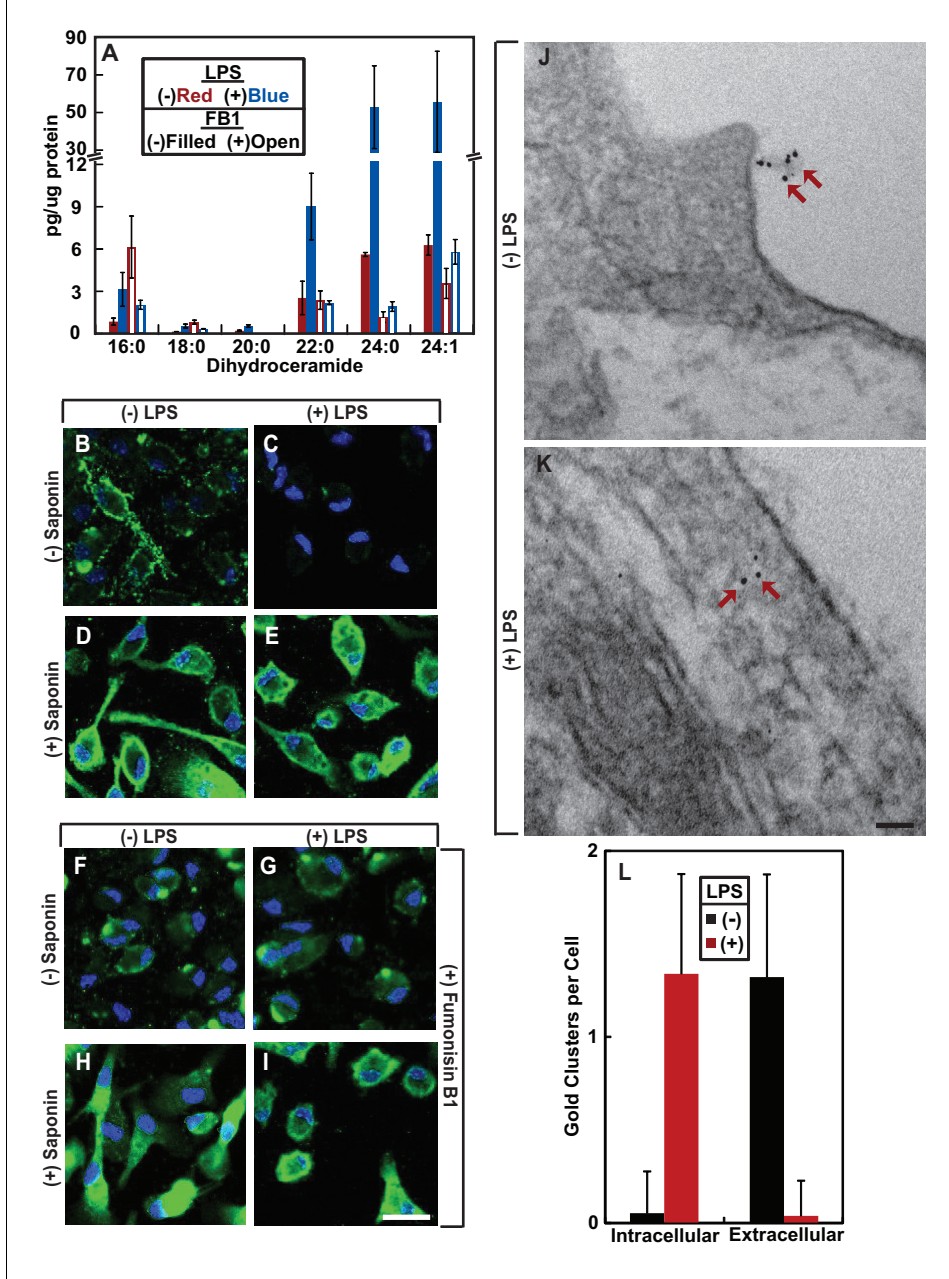

**Figure 3.** LPS induces RAT of CCR5 in primary mouse macrophages. (A–I) Mouse macrophages were treated with or without 100 ng/ml LPS in the absence (B–E) or presence of 30 µM FB1 (F–I) for 16 hr. (A) The amount of dihydroceramide with the indicated amide-linked acyl chains in the cells was determined through LC-MS measurement. Results are reported as mean ±S.E. from three independent experiments. (B–I) Macrophages were subjected to immunofluorescent microscopy analysis with an antibody against the seoncd extracellular loop of CCR5(A) in the absence or presence of saponin-mediated cell permeabilization. Scale bar = 10 µm. (J and K) Macrophages treated without (J) or with (K) 100 ng/ml LPS for 24 hr were subjected to immuno-gold EM analysis with an antibody against the N-terminal domain of CCR5. Scale bar = 200 nm. (L) The number of intracellular and extracellular-localized CCR5 labeled by gold clusters per cell was quantified from macrophages treated with (n = 20) and without LPS (n = 28). The results are reported as mean ±S.D. This number should be smaller than that of CCR5 molecules, as it did not include CCR5 labeled by a single gold particle, the specificity of which was difficult to determine.

DOI: https://doi.org/10.7554/eLife.40234.009

The following figure supplements are available for figure 3:

**Figure supplement 1.** The specificity of CCR5 immunofluorescent microscopy.

*Figure 3 continued on next page*

*Figure 3 continued*
DOI: https://doi.org/10.7554/eLife.40234.010
**Figure supplement 2.** CCR5 is localized on cell surface regardless of LPS treatment.
DOI: https://doi.org/10.7554/eLife.40234.011
**Figure supplement 3.** LPS stimulates RAT of CCR5 in macrophages.
DOI: https://doi.org/10.7554/eLife.40234.012

To rule out the possibility that the results shown above are artifacts of in vitro treatment with LPS, we injected LPS into mouse peritoneal cavity, and then isolated macrophages to determine the topology of CCR5. Since we did not treat the cells with LPS after their isolation, we analyzed the macrophages through immunofluorescent microscopy as soon as they stick to the culture plates. As a result, these cells appeared to be rounded, as they did not have time to spread out. Under this condition, LPS also induced RAT of CCR5 in macrophages, and this topological inversion was inhibited by co-injection of FB1 (*Figure 4A–L*).

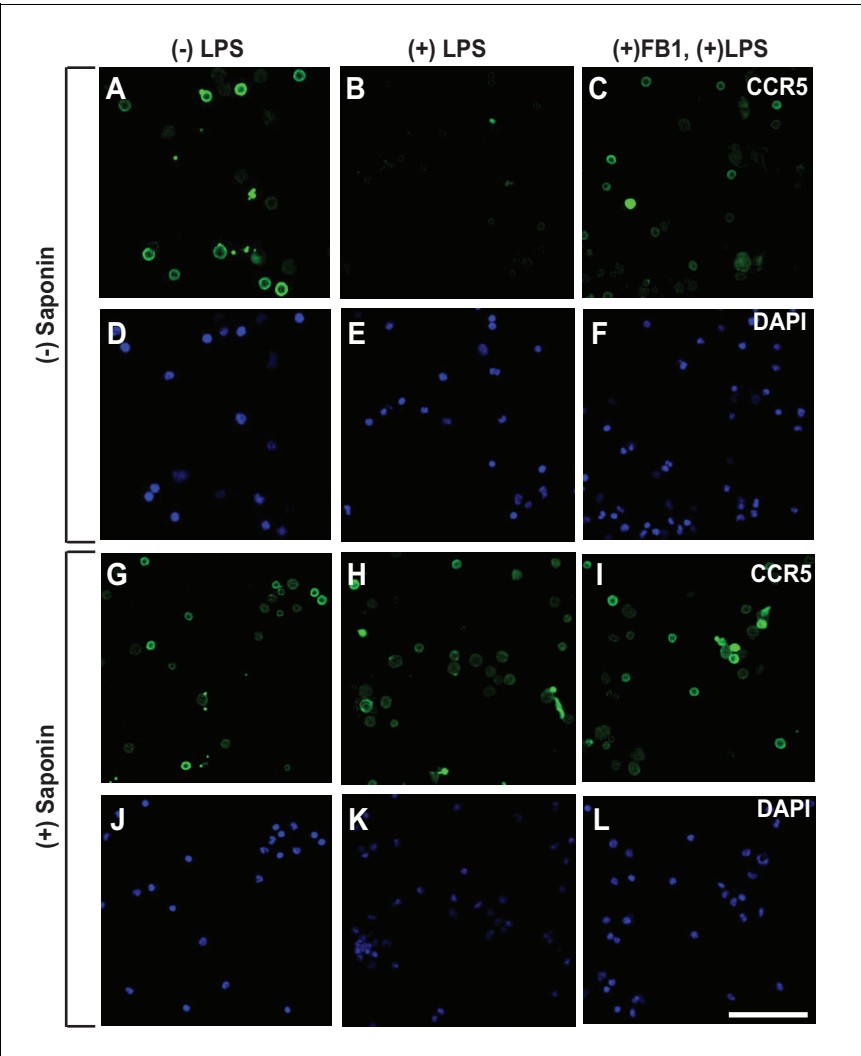

**Figure 4.** LPS injected intraperitoneally induces RAT of CCR5 in macrophages. Macrophages isolated from peritoneal cavity of mice injected intraperitoneally with 1 mg/kg LPS and 1 mg/kg FB1 for 16 hr were analyzed as described in *Figure 3B–I*. Scale bar = 20 μm.
DOI: https://doi.org/10.7554/eLife.40234.013

Since the N-terminal domain of CCR5 is in direct contact with chemokine ligands (*Duma et al., 2007*), LPS-induced topological inversion, which places this region into cytosol, should block the receptor from binding to the extracellular chemokine ligands thereby preventing macrophages from migrating toward these chemokine ligands. As expected, treatment with LPS completely blocked migration of macrophages toward CCL5 (*Figure 5A*), a known ligand for CCR5 (*Samson et al., 1997*). The migration observed in the absence of LPS was strictly dependent on CCR5, as such migration was not detected in *Ccr5*<sup>-/-</sup> macrophages (*Figure 5A*).

If LPS blocks migration of macrophages toward CCL5 by stimulating RAT of CCR5, then FB1, which prevents LPS from inducing RAT of CCR5 by inhibiting production of dihydroceramide (*Figure 3F–I*), should relieve this blockade. Indeed, co-treatment with FB1 eliminated the inhibitory effect of LPS on migration of macrophages toward CCL5 (*Figure 5B*).

## Discussion

*Figure 6* illustrates the speculative functions of RAT of CCR5. Upon bacterial infection, cells surrounding the infected tissue secrete chemokines including CCL5 (*Horuk, 2001*). In unprimed macrophages, CCL5 interacts with the GPCR configuration of CCR5 (CCR5(A)), attracting migration of these cells toward the infected sites. Upon encounter with bacterial byproducts such as LPS, the increased production of dihydroceramide in macrophages triggers RAT of CCR5, resulting in production of the protein with an inverted topology (CCR5(B)). This topological inversion blocks macrophages from further migration toward CCL5, allowing macrophages to produce pro-inflammatory cytokines to combat infection at the current location instead of migrating to the cells from which CCL5 is secreted, as these cells themselves may not be infected. On the other hand, RAT of CCR5 and other CC and CXC families of chemokine receptors containing a GXXXN motif in their first transmembrane helix may explain why macrophages overwhelmed by LPS during sepsis are markedly less effective in clearing bacterial infection, as they may be less sensitive to chemotaxis reaction (*Biswas and Lopez-Collazo, 2009*).

In addition to CCR5, CXCR4, another chemokine receptor that has been identified as a co-receptor for HIV (*Lederman et al., 2006*), also contains the GXXXN motif in the first transmembrane helix, and therefore is likely subjected to topological regulation through RAT as well. Since HIV directly interacts with the N-terminal domain and the second extracellular loop of CCR5 and CXCR4 with a topology consistent with that of GPCRs (*Dogo-Isonagie et al., 2012*; *Huang et al., 2007*; *Lu et al., 1997*), topological inversion of these receptors through RAT should block HIV entry as the regions

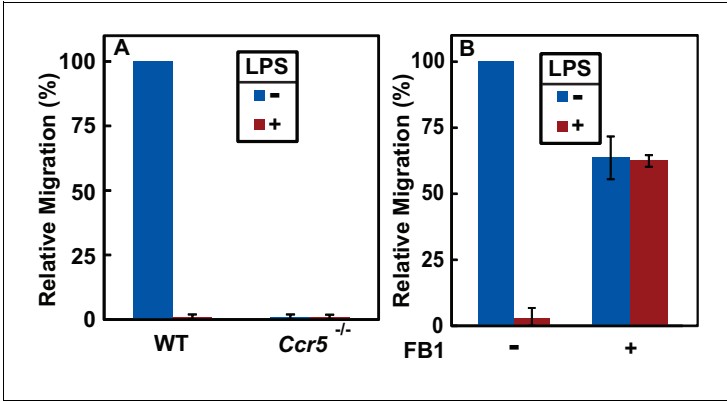

**Figure 5.** RAT of CCR5 blocks migration of macrophages toward CCL5. (**A**) Migration of macrophages from mice with indicated genotype was assayed in transwell plates for 24 hr with or without 100 ng/ml LPS in the upper wells and 100 nM CCL5 as the chemoattractant in lower wells. Signals generated in the absence of CCL5 were subtracted to normalize for random migration. (**B**) Migration assays were performed as described in (**A**) with wildtype macrophages incubated with LPS and 30 µM FB1 in upper wells as indicated. (**A and B**) Results were reported as mean ±S.E. from three independent experiments, with the value obtained from untreated WT macrophages set at 100%.

DOI: https://doi.org/10.7554/eLife.40234.014

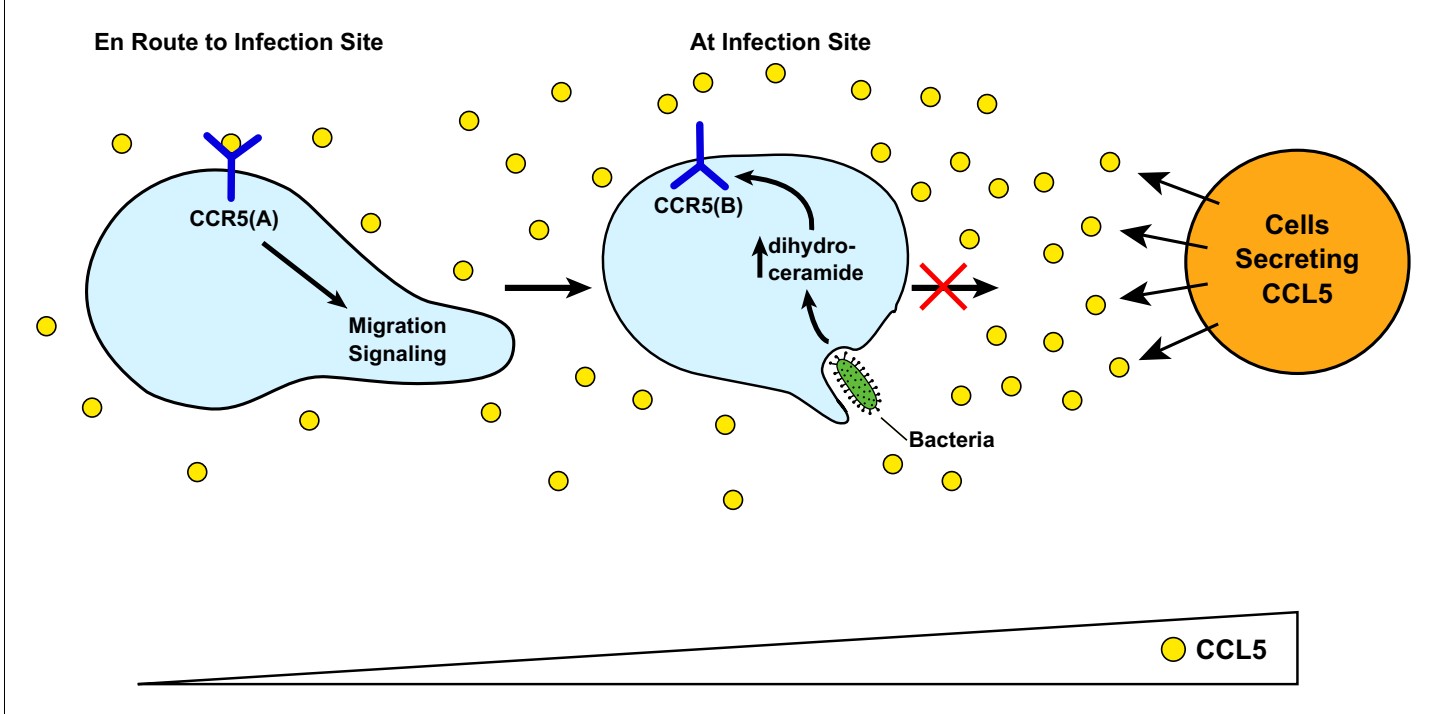

**Figure 6.** A speculative model illustrating LPS-induced topological inversion of CCR5 through RAT. Upon infection, cells surrounding the bacterial infected sites secret chemokines including CCL5. Through its interaction with CCR5 expressed in unprimed macrophages that adopts a topology consistent with that of a GPCR (CCR5(A)), CCL5 attracts migration of macrophages toward the infection site. Upon encounter with bacterial byproduct LPS, the increased production of dihydroceramide in macrophages causes RAT of CCR5, leading to expression of CCR5 with an inverted topology (CCR5(B)). This topological inversion prevents further migration of macrophages toward CCL5.
DOI: https://doi.org/10.7554/eLife.40234.015

binding the virus should be located intracellularly thereby inaccessible to the virus in circulation. This notion is consistent with previous reports showing that LPS or other treatments increasing intracellular levels of ceramide led to resistance in HIV infection (*Bernstein et al., 1991*; *Finnegan et al., 2004*; *Kornbluth et al., 1989*). These observations suggest that compounds capable of inducing RAT of these chemokine receptors but without toxicity associated with LPS may be particularly effective in combating HIV infection, as the single treatment may block both receptors from supporting entry of HIV.

The current study reveals that LPS-induced topological inversion of CCR5 may be one of the mechanisms for LPS to inhibit chemotaxis mediated by the receptor. This conclusion is supported by our observations that FB1, which blocked LPS-induced topological inversion of CCR5 by inhibiting synthesis of dihydroceramide, restored the chemotaxis reaction of macrophages exposed to LPS. Since most CC and CXC families of chemokine receptors contain a GXXXN motif in their first transmembrane helix, LPS may invert the topology of all these chemokine receptors. This scenario may explain why mice deficient in ceramide synthase 6, the major ceramide synthase expressed in macrophages, over-recruited macrophages to the inflammatory sites because of enhanced chemotaxis (*Eberle et al., 2015*), as the lack of production of dihydroceramide may not be able to inhibit these chemokine receptors through topological inversion. However, since CCR5 can be desensitized through other mechanism such as arrestin-mediated internalization (*Oppermann, 2004*), it remains unclear whether topological inversion through RAT is the only mechanism through which LPS inhibits CCR5 activity as a chemokine receptor. The best approach to test this hypothesis is to make a mutant CCR5 resisting dihydroceramide-induced RAT by locking the topology into CCR5(A). Unfortunately, fixing topology of CCR5 and TM4SF20, another protein subjected to RAT, into their A form appears to be challenging. TM4SF20 still underwent RAT when we fused a prolactin signal peptide at the N-terminus of the protein (*Chen et al., 2016*). Likewise, the N-terminally SNAP-tagged CCR5 construct used in the current study contains a signal peptide from human CD8, but the protein

still underwent ceramide-induced RAT. Thus, understanding the molecular details of the regulatory mechanism behind RAT may be required before reagents can be developed to test this hypothesis.

It should be pointed out that our data regarding inaccessibility of the antibodies against extracellular regions of CCR5(A) with the GPCR configuration in LPS or ceramide-treated cells may also be interpreted as indicating ER retention of the protein. This model may also be consistent with the observation that ceramide treatment blocked O-linked glycosylation of CCR5, as this post-translational modification takes place in the Golgi complex. However, this model cannot explain our observations that ceramide treatment exposed C-terminus of CCR5 to the extracellular space. This model is also inconsistent with the observation that significant amount of CCR5 in cells treated with ceramide or macrophages treated with LPS, which stimulated production of dihydroceramide, was localized on cell surface. Thus, taking all of our data together, we believe that topological inversion through RAT is the better interpretation of our results.

A major implication of the current study is that membrane proteins can be regulated by topological inversion through RAT. We have previously reported that ceramide stimulates RAT of TM4SF20, and this reaction depends on the presence of a GXXXN motif in the first transmembrane helix of the protein (*Chen et al., 2016*). In the present study we identified ~100 proteins containing this motif in their first transmembrane helix, and demonstrated that CCR5, one of the proteins in the list, can indeed be regulated through RAT. Interestingly, most proteins in the list including CCR5 are GPCRs. In order for these proteins to function as GPCRs, the N-terminal end of the first transmembrane helix has to insert into ER lumen (*Pierce et al., 2002*). However, vast majority of GPCRs do not contain an N-terminal cleavable signal peptide (*Guan et al., 1992*), the well-characterized mechanism directing N-terminus of a protein into ER lumen (*Zimmermann et al., 2011*). Our finding suggests that the presence of a GXXXN motif within the first transmembrane helix of many GPCRs may be a previously unrecognized mechanism for the transmembrane helix to be inserted with such an orientation. Moreover, the presence of this motif may allow these GPCRs to be regulated through topological inversion upon accumulation of dihydroceramide, ceramide or other related sphingolipids. Such regulation not only inhibits functions of the GPCR, but also simultaneously activates functions performed by the protein with the inversed topology. Proteins sharing the architecture of GPCRs that contain seven transmembrane helices but adopting an inverted topology have been reported to function as receptors independent of G proteins (*Deckert et al., 2006*) and ion channels (*Sato et al., 2008*). Delineating functions of GPCRs with the inverted topology will greatly enhance our understanding of these receptors.

An effective approach to delineate the function of CCR5 with the inverted topology is to make a mutant CCR5 that is locked into CCR5(B) regardless of the presence of ceramide or dihydroceramide. We previously reported that mutating the glycine or asparagine residue to leucine in the GXXXN motif in the first transmembrane helix of TM4SF20 locked the topology of the protein into the B form, and the mutant protein performed a function similar to that of wild type TM4SF20 with the inverted topology (TM4SF20(B)) (*Chen et al., 2016*). However, when we made a similar mutation in CCR5 (CCR5(N48L)), we observed that unlike wild type CCR5 with the inverted topology (CCR5(B)) that is capable of reaching cell surface, the mutant protein was exclusively localized in the ER. As a result, we are unable to rule out the possibility that the N48L substitution affects proper folding of the protein, making this model unsuitable to study the function of CCR5 with the inverted topology. Thus, delineating the molecular mechanism behind RAT may be required before reagents can be developed to determine the functions of CCR5 with the inverted topology.

## Materials and methods

**Key resources table**

| Reagent type (species) or resource | Designation | Source or reference | Identifiers | Additional information |
|---|---|---|---|---|
| Gene () | NA | NA | | |

*Continued on next page*

*Continued*

| Reagent type (species) or resource | Designation | Source or reference | Identifiers | Additional information |
|---|---|---|---|---|
| Strain, strain background () | C57Bl/6 (Mus musculus, male and female) | UTSW Breeding Core | NA | |
| Strain, strain background () | *Ccr5⁻/⁻* (Mouse, male and female) | Jackson Laboratories | Stock #: 005427 | |
| Genetic reagent () | NA | | | |
| Cell line () | HEK-293 cells (Homo Sapiens) | ATCC | ATCC CRL-3216 | |
| Cell line () | SV589 cells (human) | NIGMS Human Genetic Mutant Cell Repository | NA | Discontinued for distribution |
| Transfected construct () | p*CCR5-Myc* (human) | This paper | NCBI Reference Sequence: NM_000579.3 | Encodes full length human CCR5 followed by five tandem repeats of the Myc epitope tag. |
| Transfected construct () | p*β2AR-SNAP*; p*CCR1-SNAP*; p*CCR4-SNAP*; p*CCR5-SNAP*; p*CCR10-SNAP*; and p*MAS1-SNAP* (human) | This paper | NCBI Reference Sequence for β2AR, CCR1, CCR4, CCR5, CCR10 and MAS1 are NM_000024.5, NM_001295.3, NM_005508.4, NM_000579.3, NM_016602.3 and NM_002377.3, respectively. | Encode indicated full length human GPCRs followed by a C-terminal SNAP-tag |
| Transfected construct () | p*SNAP-CCR5* (human) | Cisbio | Cat#PSNAPCCR5 | |
| Biological sample () | NA | | | |
| Antibody | IgG-9E10 | ATCC | ATCC CRL-1729 | 0.5 µg/ml for immunoblot analysis, 3 µg/ml for immunofluorescent microscopy |
| Antibody | Human CCR5 Antibody, 45531 | R and D Systems | Cat#MAB182-100 | 1 µg/ml |
| Antibody | Actin Antibody | Sigma Aldrich | Cat#A2066-100UL | 1:10,000 dilution |
| Antibody | Anti-SNAP-tag Antibody | New England Biolabs | Cat#P9310S | 1:1000 dilution |
| Recombinant DNA reagent | NA | | | |
| Sequence-based reagent | NA | | | |
| Peptide, recombinant protein | Recombinant Human CCL5/ RANTES Protein | R and D Systems | Cat#278-RN-050 | |
| Commercial assay or kit | CLIP-Surface Starter Kit | New England Biolabs | Cat#E9230S | |

*Continued on next page*

*Continued*

| Reagent type (species) or resource | Designation | Source or reference | Identifiers | Additional information |
|---|---|---|---|---|
| Commercial assay or kit | SNAP-Lumi4-Tb | Cisbio | Cat#SSNPTBC | |
| Commercial assay or kit | Pierce Cell Surface Protein Isolation Kit | Thermo Fischer Scientific | Cat#89881 | |
| Commercial assay or kit | CytoSelect 96-Well Cell Migration Assay | Cell Biolabs, Inc. | CBA-105 | |
| Chemical compound, drug | Xtreme Gene HP DNA Transfection Reagent | Sigma Aldrich | Cat#6366244001 | |
| Chemical compound, drug | Saponin from quillaja bark | Sigma Aldrich | Cat#S4521-25G | |
| Chemical compound, drug | Lipopolysaccharides from Escherichia coli 0111:B4 | Sigma Aldrich | Cat#L3024-5MG | |
| Chemical compound, drug | Fumonisin B1 from Fusarium moniliforme | Sigma Aldrich | Cat#F1147-1MG | |
| Chemical compound, drug | N-Hexanoyl-D-sphingosine ($C_6$-Ceramide) | Sigma Aldrich | Cat#H6524-1MG | |
| Software, algorithm | BLAST, blastp suite | NCBI | NA | |
| Other | NA | | | |

## Materials

We obtained anti-human CCR5 45531 from R and D Systems (Minneapolis, MN), anti-actin from Sigma Aldrich (St. Louis, MO), anti-Giantin 924302 from Biolegend (San Diego, CA), Alexa Fluor 488 FluoroNanogold goat anti-mouse IgG Fab from Nanoprobes.com (Yaphank, NY), AffiniPure Donkey Anti-Rabbit IgG (H + L) from Jackson ImmunoResearch (West Grove, PA), Alexa Fluor 488 goat Anti-Mouse IgG (H + L) from Invitrogen (Carlsbad, CA), Anti-SNAP-tag Antibody (Polyclonal) from New England Biolabs (Ipswichm, MA). Hybridoma cells expressing anti-Myc 9E10 were obtained from ATCC. Saponin (from quillaja bark), LPS (from Escherichia coli 0111:B4), glutaraldehyde, and fumonisin B1 (from Fusarium moniliforme) was purchased from Sigma Aldrich (St. Louis, MO). Recombinant human CCL5 was obtained from R and D Systems (Minneapolis, MN).

## Mice

Male and female littermates of 6–8 week-old mice of C57Bl/6 background were used for all studies under APN# 2015–100860 approved by UTSW IACUC. Wildtype mice were ordered from UTSW Breeding Core. *Ccr5*$^{-/-}$ mice (Stock: 005427) were purchased from Jackson Laboratories (Bar Harbor, ME).

## Cells

HEK293 (human female embryonic kidney cells) and SV589 (human male transformed fibroblasts) cells were maintained in medium A (Dulbecco's modified Eagle's medium with 4.5 g/l glucose, 100 U/ml penicillin, 100 µg/ml streptomycin sulfate, and 5% fetal calf serum) in monolayers at 37°C in 8% and 5% $CO_2$, respectively. To guard against potential genomic instability, an aliquot of each cell line is passaged for only 4 weeks before a fresh batch of cells is thawed and propagated for experimental use. All the cell lines have been confirmed to be free of mycoplasma infection using the MycoAlert Mycoplasma Detection Kit (Lonza, Allendale, NJ).

To obtain primary mouse macrophages, mice were intraperitoneally injected with 1 ml of 38.5 mg/ml thioglycolate. After 4 days, 3 ml phosphate buffer saline (PBS) was injected into abdomen of the mice euthanized through isoflurane overdosing. After brief massage, cells suspended in PBS were extracted and seeded in medium B (Dulbecco's modified Eagle's medium with 4.5 g/l glucose, 100 U/ml penicillin, 100 µg/ml streptomycin sulfate, and 10% fetal calf serum) at 37°C in 5% $CO_2$. After 2 hr, non-macrophage cells were removed by multiple washes of medium B. Primary macrophages, which stick to the plates, were cultured medium B in monolayers at 37°C in 5% $CO_2$.

## Plasmids

The original cDNA clone for human CCR5 was obtained from UTSW Vector Core Laboratory (IOH27324). pCCR5-Myc encodes full length human CCR5 followed by five tandem repeats of the Myc epitope tag. pSNAP-CCR5 was purchased from CisBio and encodes full length human CCR5 preceded with an N-terminal SNAP-tag. pβ2AR-SNAP, pCCR1-SNAP, pCCR4-SNAP, pCCR5-SNAP, pCCR10-SNAP and pMAS1-SNAP encode indicated full length human GPCRs followed by a C-terminal SNAP-tag. CCR5 mutants were generated through site-directed mutagenesis with the Quik-Change Multi Site-Directed Mutagenesis Kit (Agilent Technologies, Santa Clara, CA) on plasmids encoding CCR5 described above. Desired mutations were confirmed by sequencing the entire open reading frame of the gene.

## Immunoblot analysis

On day 0, SV589 cells were seeded at $5 \times 10^5$ cells per 60 mm dish. On day 1, cells were transfected with 0.15 µg/dish of Myc-tagged WT or mutant CCR5 plasmid. Following treatments described in the figure legends, cells were lysed in buffer A (25 mM Tris-HCl, pH 7.2, 150 mM NaCl, 1% NP40, 0.5% Sodium Deoxycholate, 0.1% SDS) containing cOmplete Protease Inhibitor cocktail (Roche, Indianapolis, Indiana). After brief centrifugation, buffer B (62.5 mM Tri-HCl, pH6.8, 15% SDS, 8 M Urea, 10% glycerol, 100 mM DTT) was added to clarify lysate at 1:1 ratio. Cell lysate was analyzed by SDS-PAGE followed by immunoblot analysis with the indicated antibodies (1:2000 dilution for anti-Myc, 1:10,000 dilution for anti-actin). Bound antibodies were visualized with a peroxidase-conjugated secondary antibody using the SuperSignal ECL-HRP substrate system (Pierce, Waltham, Massachusetts).

## SNAP-tag fluorescent labeling

On day 0, HEK293 cells were seeded at $5 \times 10^5$ cells per 60 mm dish. On day 1, cells were transfected with 0.5 µg/dish of SNAP-tagged GPCRs. Following treatments described in the figure legends, for experiments shown in *Figure 1B* cells were treated with 5 µM cell-impermeable (SNAP-Surface 488) or cell–permeable (SNAP-Cell 505 Star) SNAP-tag substrate. For experiments shown in *Figure 2D*, cells were treated with 100 nM cell-impermeable substrate Lumi4-Tb (CisBio, Bedford, MA). The labeling reaction was carried out for 1 hr at 37°C in 8% $CO_2$. Labeled cells were washed in Tag-Lite Buffer (CisBio) and resuspended to $7.5 \times 10^5$ cells/ml. Cell suspension (100 µl) were added in triplicate to a 96-well dish and fluorescence was measured using a Tecan plate reader with excitation and emission of 506 and 526 nm for SNAP-Surface 488 and SNAP-Cell 505 Star or an excitation and emission of 340 and 620 nm for Lumi4-Tb, respectively.

## Biotin labeling of cell-surface cysteine residues

On day 0, SV589 cells were seeded at $5 \times 10^5$ cells per 60 mm dish. On day 1, cells were transfected with 0.15 µg/dish of Myc-tagged WT or mutant CCR5 plasmid. Following treatments described in the figure legends, cell surface cysteine residues were labeled by 0.25 mg/ml EZ-link maleimide-PEG2-biotin (Thermo Fisher Scientific) and immunoprecipitated with the Pierce Cell Surface Labeling Kit according to manufacturer's direction. Resulting fractions were mixed with buffer B at 1:1 ratio and subjected to immunoblot analysis.

## Immunofluorescent microscopy and immuno-gold EM analysis of CCR5

For immunofluorescent microscopy of transfected SV589 cells, $2 \times 10^5$ cells were seeded on each 35 mm Magtek plate on day 0. On day 1, cells were transfected with the indicated plasmids (0.1 µg/dish). After 2 hr, cells were treated with or without $C_6$-ceramide. 16 hr later on day 2, cells were fixed in DMEM containing 4% PFA for 5 m, washed with PBS, and then incubated in the absence or

presence of 0.25% saponin in PBS for 10 m as indicated. After a PBS wash and blocking with Mouse On Mouse (MOM, Vector Laboratory, Burlingame, CA) blocking buffer for 30 m, cells were incubated with 3 µg/ml 9E10 antibody or anti-Giantin (1:600 dilution) diluted in MOM Diluent buffer for 40 m. Following another wash, cells were stained with 4 µg/ml Alexa Fluor 488 goat anti-mouse IgG, 4 µg/ml Alexa Fluor 633 goat anti-rabbit IgG and 0.1 µg/ml DAPI diluted in MOM diluent buffer for 1 hr. The cells were then washed and subjected to confocal imaging analysis using the Zeiss LSM880 Airyscan microscope utilizing the Zen Digital Imaging Software.

For immunofluorescent microscopy of Primary mouse macrophages, cells were set up at $1 \times 10^6$ cells per 35 mm No. 1.5 coverslip dish, and treated as described in the figure legend. Plates containing mouse primary macrophages were washed with PBS, fixed in PBS containing 1% glutaraldehyde and 4% PFA for 20 min, washed with PBS, and incubated with 50 mM glycine dissolved in PBS for 15 min to block the remaining glutaraldehyde. For cells subjected to permeabilization, the plates were treated with 0.25% saponin in PBS for 30 min. Following this step the wash was performed with either PBS (without cell permeabilization) or PBS containing 0.2% saponin (with cell permeabilization). After a wash, plates were blocked with MOM blocking buffer for 1 hr. Plates were then incubated with 1 µg/ml 45531 antibody diluted in MOM Diluent buffer overnight at 4°C. Following another wash, plates were incubated with 4 µg/ml Alexa Fluor 488 goat anti-mouse IgG and 0.1 µg/ml DAPI diluted in MOM diluent buffer for 2 hr. The plates were then washed and subjected to confocal imaging analysis using the Zeiss LSM880 Airyscan microscope utilizing the Zen Digital Imaging Software.

For immuno-gold EM analysis, the plates were treated the same way as that of immunofluorescent microscopy for permeabilized cells up to the step before addition of the secondary antibody. For this purpose, the plates were incubated with 10 µg/ml Alexa Fluor 488 FluoroNanogold goat anti-mouse IgG for 2 hr. The plates were then washed with PBS containing 50 mM glycine, and submitted to the UTSW EM Core for gold enhancement, sectioning, and sample preparation. Imaging was performed on a Tecnai Spirit electron microscopy utilizing the iTEM software and a Morada camera.

## Migration assay

Primary mouse macrophages from mice were seeded into upper wells of transwell plates at $2 \times 10^5$ cells per well, and treated as described in the figure legend. Migration assays were performed with the CytoSelect 96-Well Cell Migration Assay (Cell Biolabs, Inc., San Diego, CA) using Corning HTS Transwell 96 well permeable supports (Sigma-Aldrich, St. Louis, MO) according manufacturer's direction. Signals generated in the absence of CCL5 were subtracted to normalize for random migration.

## Sphingolipid quantitation

Primary mouse macrophages were set up at $5 \times 10^6$ cells per 100 mm dish, and treated with or without 100 ng/ml LPS for 24 hr. Sphingolipids including dihydroceramide were quantitated by the UT Southwestern Lipidomics Core as previously described (*Denard et al., 2012*)

## Bioinformatics analysis

Pattern Hit Initiated BLAST was performed using the first transmembrane domain of TM4SF20 as the query sequence combined with the following PHI-BLAST algorithm: [LVAWPMIF](2-8)GXXXN [LVAWPMIF](5-7). Resulting proteins confirmed to have the GXXXN motif within the first transmembrane domain were included to run three additional rounds of Position-Specific Iterative BLAST. All proteins reported in *Table 1—source data 1* were checked manually to confirm that they contain the GXXXN motif in the first transmembrane helix.

## Statistical analysis

Methods of quantification and statistical analysis are reported in the Figure Legends.

## Acknowledgments

We thank Joachim Seemann, Drs. Brown and Goldstein for their helpful comments and constant support; Lisa Beatty, Ijeoma Dukes, Nimisha Jacob and Lauren Valsin for help with tissue culture; Elina

Esmaeilzadeh for technical assistance; and Nancy Heard and Chelsea Burroughs for graphic illustration. We acknowledge UTSW EM Core, Live Cell Imaging Core, and Metabolic Phenotyping Core for their technical support. This work was supported by grants from the National Institutes of Health (GM-116106 and HL-20948) and Welch Foundation (I-1832).

## Additional information

### Funding

| Funder | Grant reference number | Author |
|---|---|---|
| National Institute of General Medical Sciences | GM-116106 | Jin Ye |
| National Heart, Lung, and Blood Institute | HL-20948 | Jin Ye |
| Welch Foundation | I-1832 | Jin Ye |

The funders had no role in study design, data collection and interpretation, or the decision to submit the work for publication.

### Author contributions

Bray Denard, Conceptualization, Data curation, Formal analysis, Investigation, Methodology, Writing—original draft; Sungwon Han, Data curation, Validation, Investigation, Methodology; JungYeon Kim, Resources, Data curation; Elliott M Ross, Resources, Formal analysis, Writing—review and editing; Jin Ye, Conceptualization, Resources, Formal analysis, Supervision, Funding acquisition, Investigation, Methodology, Writing—original draft, Project administration, Writing—review and editing

### Author ORCIDs

Sungwon Han (iD) http://orcid.org/0000-0002-4097-856X
Jin Ye (iD) http://orcid.org/0000-0002-3179-4777

### Ethics

Animal experimentation: Male and female littermates of 6-8 week-old mice of C57Bl/6 background were used for all studies under APN# 2015-100860 approved by UTSW IACUC.

### Decision letter and Author response

Decision letter https://doi.org/10.7554/eLife.40234.018
Author response https://doi.org/10.7554/eLife.40234.019

## Additional files

### Supplementary files

• Transparent reporting form
DOI: https://doi.org/10.7554/eLife.40234.016

### Data availability

All data generated or analysed during this study are included in the manuscript and supporting files.

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
