## [Decision Letter]

[**Editorial note:** This article has been through an editorial process in which the authors decide how to respond to the issues raised during peer review. The Reviewing Editor's assessment is that major issues remain unresolved.]

Evaluation of the revised submission:

We reiterate that there is potential to test your hypothesis further for chemokine receptors CXCR1 (SLLGN) and CXCR2 (SLLGN), which have a first helix lacking the GXXXN motif, and which are more similar b2 adrenergic receptor (IVFGN), which did not display the signatures of inversion. It would be very interesting to learn if these receptors would continue to function after strong LPS signaling, suggesting that cells expressing these receptors (including neutrophils) might still respond to endogenous chemokines in the context of bacterial sepsis.

Reviewer #2:

I have relatively little to add to my original concerns. Readers of the transparent review process can judge for themselves how convincing the authors' conclusions are. I would re-iterate the following issues which remain particularly problematic in my view:

1) In response to my original point (3), the authors have simply removed all data regarding the G44L and N48L mutants because they could not be used to support their model. This substantially weakens the paper in two ways (and calls into question their interpretation of analogous mutants analyzed in their earlier paper). First, the notion that this motif is a key determinant of RAT simply cannot be supported because there are now no data provided in the manuscript to probe this. Second, they now have no experiments that actually perturb RAT by manipulating translocation to convince a reader that the effects can indeed be traced to altered protein topogenesis.

2) The authors' response to my original point (4) is not consistent with a rather large body of data on how protein translocation works. I had pointed out that if their SNAP-CCR5 construct contains a signal peptide at the N-terminus, the signal would enforce translocation of the N-terminus into the lumen thereby ensuring the canonical CCR5 topology. The authors countered that the downstream TMD "prevails over...the signal peptide". This might be feasible if the sequence between the signal and TMD was relatively short and unstructured; however, the SNAP tag is neither short (being ~185 amino acids long) nor unstructured. Because topogenesis occur co-translationally (very well established), the signal peptide will initiate translocation long before the TMD is even synthesized (also very well established). By the time the TMD emerges from the ribosome, the entire SNAP tag (~20 kD) will be in the lumen. It is exceedingly unlikely that the TMD can 'override' this situation and somehow cause the lumenal SNAP tag to be retrieved to the cytosol. If this is indeed the claim, the available data is simply too weak to overturn (or ignore) the considerable literature on this issue. For this reason, I feel the authors are almost certainly wrong in their interpretation. Because they nonetheless see RAT by their assay, I would strongly suspect a flaw in their assay rather than ignoring well-established principles of how signal peptides and protein translocation work.

3) The authors' response to point (6) actually argues against them. I had pointed out that immunogold EM labeling has a spatial precision of at best 20-30 nm, which would pose a challenge for determining whether an epitope was on one or the other side of a 5 nm thick membrane. I also noted that some of their gold clusters were ~200 nm away from the plasma membrane, making me think they were observing intracellular antigens in these cases. The authors countered that they felt an antibody sandwich could indeed reach 200 nm away (how is unclear given the known dimensions of an antibody). Regardless, if one does believe the precision is 200 nm, it is an even worse technique for determining topology where 5 nm resolution is needed. The authors cite earlier studies using immunogold EM for topology, but previous ill-advised uses of a method is not a strong argument for using it again now.

Minor Comments:

1) The fluorescence micrographs remain of poor resolution and one cannot judge in most cases whether the staining actually represents surface localization as claimed (e.g., Figure 1 and Figure 2–figure supplement 1).

2) The authors claim that gold clusters are sometimes almost 200 nm away from the surface on the extracellular side as well. However, in each instance shown in the paper, these clusters seem to have membrane patches under them.

*Reviewer #3:*

The authors have added some sentences to the discussion and modified Figure 6 (summary) but in the end really did not address the questions I asked.

The manuscript suggests that chemokine receptors could be regulated by inversion. If that were the case than as a cell is migrating at some point more inversion would occur to allow it to stop at its destination. Adding LPS to the system simply complicates the model and may represent a hijacking of the inversion by bacterial products. It would be nice to know whether inversion ever blocks a chemotaxing cell without having to invoke LPS. In figure 5B there is some suggestion this may be the case as fewer cells are migrating when FB1 is added to chemotaxing cells (about 35% inhibition).

My point is that the paper as it stands is not really about chemotaxis but about how LPS might block chemotaxis. It’s really the pathology of a system for which the physiology has not been worked out. Does inversion play a role under normal chemotaxis to stop cells? I think this would be important for this paper.

Additional data files and statistical comments:

My comment is the same as in my first review. What is the physiologic importance of inversion of a chemokine receptor?

Decision letter after peer review:

Thank you for submitting your article "Regulating G Protein-Coupled Receptors by Topological Inversion" for consideration by *eLife*. Your article has been reviewed by three peer reviewers, including Michael L Dustin as the Reviewing Editor and Reviewer #1, and the evaluation has been overseen by Richard Aldrich as the Senior Editor. The other two reviewers remain anonymous.

The Reviewing Editor has highlighted the concerns that require revision and/or responses, and we have included the separate reviews below for your consideration. If you have any questions, please do not hesitate to contact us.

Currently there are some major issues to resolve. This assessment is based on the reviews and our subsequent discussion in which the reviewers could see each other comments. Reviewer 1 (MLD) and 3 were largely accepting of the cell biology, but had questions about the biological implications. Reviewer 2 has a number of technical issues regarding the demonstration of inversion of the chemokine receptor rather than other mechanisms, such as ER retention. To provide a little more context, reviewer 1 considered points 2 and 6 of reviewer 2 at length and decided that your interpretation was consistent with the data. But reviewer 2 makes other points that would leave major issues unresolved with the paper in its current form.

Separate reviews (please respond to each point):

*Reviewer #1:*

The work is technically sound and I have no concerns with the major conclusions regarding the time frame of topological inversion and the loss of sensitivity to CCR5 ligands. Prior work has demonstrated a rapid p38, and recently, arrestin dependent suppression of responses to classical chemokine and leukotriene binding GPCR by receptor for chemoattractants that more directly identify the location of pathogens like the fMLP receptor or C5a receptor. This hierarchy is thought to help direct leukocytes into tissues through the chemokine and leukotriene receptors and then allow local prioritisation of signals that best allow phagocytes to target microbes. The mechanism discovered here develops over several hours, but seems to eliminate function of most of the CC and CXC chemokine receptors (except CXCR1 and CXCR2 by sequence) and the microbe associated fMLP and C5a receptors, which also have the GXXXN motif in helix 1 and may be blinded by this mechanism. As reagents may be available to more broadly investigate inversion of GXXXN vs non-GXXXN GPCR expressed by phagocytes in response to LPS could shed light on the extent to which the sequence is actually predictive of this phenomenon. The authors have raised the issue by drawing in their interpretive drawing that this mechanism also allows phagocytes to locate microbes, but based on the sequence information, this doesn't seem to be the case and the LPS treated phagocytes will also be blinded to microbes after about 16 hours. This would give this mechanism a broader impact on leukocyte chemotaxis than the previously defined hierarchies. This would seem to have implications for challenges with clearing extracellular microbes in sepsis.

*Reviewer #2:*

This manuscript arrives at the remarkable conclusion that CCR5 (and certain other GPCRs) can be made in a fully inverted topology in response to altered ceramide levels. The functional consequence of this inversion is the reduced biological activity of the receptor, which in the case of CCR5 involves macrophage migration toward its ligand, CCL5. Due to the striking nature of these claims, we read the manuscript with great interest. While there is no doubt that ceramide affects CCR5 in some manner, numerous conceptual and experimental flaws in the study's design and interpretation greatly weaken the argument that the basis of these effects is topology inversion. When the below points are considered together, it would seem that much of the data argue for ceramide affecting CCR5 trafficking, not topology. I would therefore strongly encourage the authors to reconsider publication of this study until the data for an effect on topogenesis is examined more carefully and thoroughly. Indeed, it is noteworthy (and rather surprising) that not a single experiment in the paper actually looks at CCR5 topogenesis at the ER, opting instead for relatively blunt end-point assays prone to alternative interpretations as outlined below.

Major concerns:

1) The assays in which the C-terminus of CCR5 is tagged and monitored by cell impermeable labelling are only sufficient to qualitatively demonstrate the topological location of the tagged terminus, but cannot accurately report on CCR5 topogenesis as a whole. It is easily conceivable that drastic changes in lipid composition within the cell result in misfolding or incomplete insertion of some TMDs such that the C-terminus winds up in the ER lumen at some low level. Given that these are all performed with an overexpressed protein, it is also conceivable that a misfolded population would "escape" the quality control machinery and successfully reach the PM. For these reasons, it is important to carefully quantify the amount of surface-exposed 'inverted' CCR5 relative to total cellular CCR5. One way to do this would be to use biotin-SNAP to label the surface exposed inverted CCR5, then use avidin pulldowns and blots to determine what proportion of total cellular CCR5 was surface labeled. Controls using pre-labeling with fluorescent SNAP label prior to biotin labeling could verify that labeling efficiencies are close to quantitative. Such an experiment would also verify that what is being detected is actually full length CCR5, not a fragment.

2) In several of the critical experiments aimed at demonstrating topology inversion, the authors use glycosylation as an assay for topology. While N-linked glycosylation is a reasonable proxy for topology, O-linked glycosyation is problematic. The reason is that this reaction occurs in the Golgi on the folded protein. Thus, the absence of O-linked glycosylation could be due to a failure of the protein to reach the Golgi and/or altered folding, and not necessarily indicative of topology. Furthermore, I did not see where the authors provided evidence that O-linked glycosylation is not impacted by 16 h of ceramide treatment. This would seem to be a critical control for any assay relying on an endogenous biochemical reaction.

3) G44L and N48L are used as mutants intended to confine topogenesis to the inverted form. However, these mutants are problematic for three reasons. First, inspection of the crystal structure of CCR5 reveals that they both face the interior of the folded protein, with N48 interacting with D76, and the absence of a side chain on G44 being important for avoiding clashes with L80. Thus, mutating either residue to Leucine will very likely disrupt the folding of the protein, and hence might cause it to be primarily retained in the ER. This would readily explain why it is not O-glycosylated by Golgi enzymes (Figure 2B) and why none of it is labelled at the surface in Figure 2E without having to invoke an inverted topology. Second, the non-conservative mutations that substantially increase hydrophobicity is ill-advised as it may have effects on topology independently of the GXXXN motif. Instead, mutating this motif to residues of comparable hydrophobicity is preferable to more cleanly illustrate that N is crucial to ceramine-induced effects. Even this may well be confounded by effects on protein structure/stability. Third, the increased hydrophobicity of the G44L or N48L mutations is actually predicted to favour the correct orientation with the N-terminus facing the extracellular side (see PMID 9151664), not the inverted orientation as proposed by the authors. For all of these reasons, it is quite unlikely that mutants are acting in the way the authors propose.

4) The SNAP-CCR5 construct used in Figure 2E and Figure 2—figure supplement 2 would seem to directly disprove the authors' conclusions. As stated in the main text, the construct contains a SNAP tag (~20 kD) at the N-terminus, and according to the Materials and methods, this was obtained from CisBio. The sequence of the construct from CisBio indicates that the construct contains the N-terminal signal sequence from human CD8. This would explain how the SNAP tag domain actually gets translocated across the ER membrane. Thus, the topology of this construct is determined by the N-terminal signal sequence, not the first transmembrane domain as in the untagged construct. Rather remarkably, the SNAP-CCR5 also shows strong ceramide-induced exposure at the cell surface (Figure 2E) and loss of O-linked glycosylation (supplement). Since this construct's topology is not determined by the first transmembrane domain, it would seem that both the surface exposure and glycosylation are affected by ceramide for other reasons. The simplest interpretation is retention of the protein in the ER after ceramide treatment, which would also explain why the N48L is constitutively not only the surface and why it is constitutively not glycosylated. The authors seems to have completely overlooked the fact that this construct has a signal sequence. If they have removed it, then it is not stated in the Materials and methods (and it would be very hard to explain how the N-terminus could be translocated across the ER membrane).

5) The cysteine mutations intended to be a reporter of topology are problematic because several cysteines on the cytosolic side of CCR5 are palmitoylated. Thus, mutating these may well have consequences for trafficking of the receptor (see PMID 21819967 for a review), confounding any interpretations from this approach.

6) The authors use immunogold labelling to infer topology of CCR5 in macrophages (Figure 3J-3L). This is conceptually problematic and the observations do not seem to fit with their favoured interpretation. Given that their primary antibody recognizes an epitope close to the membrane and the secondary antibody appears to be a gold-conjugated Fab (not entirely clear in the Materials and methods), the label can be up to ~21 nm away from the epitope (~14 nm for an IgG, and ~7 nm for the Fab). This distance increases to ~28-30 nm if the secondary antibody is a whole IgG. In either case, the label can easily be on either side of the membrane regardless of the topology because the membrane thickness is only ~5 nm. Hence, without extensive numbers and statistical power, one simply cannot get the 5 nm resolution needed to infer topology using a labelling method with a precision of 20-30 nm. Remarkably however, they do see a difference. But inspection of the micrograph in Figure 3K shows that the gold particles are ~200 nm or more away from the plasma membrane. As should be obvious from the above considerations, this cannot be compatible with labelling of a cell surface protein. Many of the examples in the supplement are also too far away (although a scale bar was not provided). Thus, the simplest explanation is that after LPS treatment, the CCR5 that is labelled is in intracellular compartments, perhaps the ER (which is often close to the plasma membrane). However, the membrane morphology is not retained very well in their EM images, so one cannot really evaluate this. A second major issue is that they only observe an average of ~1.3 labels per cell. It is difficult to know what to make of this given how much signal is seen in the fluorescent images using the same antibody. For these reasons, the authors' interpretation of the immuno-EM cannot be supported from the data they present.

7) In the absence of the immuno-EM experiment, the macrophage data is very weak in establishing any role for topology inversion of CCR5 in LPS-stimulated macrophage migration. Figure 3A to 3I simply establishes that LPS reduces surface levels of CCR5 and does not speak at all to its topology. The same is true for Figure 4. All of these data are equally consistent with LPS causing intracellular retention (or endocytosis) of CCR5 in a dihydroceramide dependent manner. Thus, the effects on migration seen in Figure 5 simply reflects the fact that there is less CCR5 on the surface of these macrophages.

Minor Comments:

1) The fluorescence microscopy is poor resolution and often over-exposed (at least in the images I was provided) so it is very hard to judge localization (surface versus intracellular locations).

2) Introduction – the authors state that "it remains unclear how GPCRs can adopt such a membrane topology without the signal peptide." This is not accurate, as the study of how an N-terminal transmembrane domain directs topology has been extensively studied. Such sequences are termed signal anchors because they act as both a signal sequence and transmembrane anchor. The features that determine their topology has been investigated by extensive mutagenesis many years ago (see PMID 15461443 for a review, and PMIDs 9151664, 8557050, and 1985975 for some of the primary papers), and quite a bit has also been done on the molecular basis of their insertion (see PMID 10943843).

3) Throughout the paper, the authors essentially assume that an inverted GPCR would traffic to the cell surface. This seems very unlikely because the extracellular domain facing the cytosol would not form disulphide bonds and therefore not fold correctly, while the C-terminus facing the outside would not be palmitoylated. A more nuanced discussion is merited.

4) The surface labelling kit cited in the Materials and methods labels amines, not sulfhydryls. Some clarification is needed here about how the experiment was actually done.

*Reviewer #3:*

This study proposes a very nice mechanism of chemokine receptor inversion to explain why cells may stop chemotaxing. While interesting, the biological situation chosen is a difficult one to understand i.e., LPS stimulation of peritoneal macrophages prevents migration to CCL5. I will ask a few questions that would need to be answered with additional experiments.

There is much work to demonstrate that when high enough concentrations of a chemokine for sufficient time leads to downregulation of the receptor or the receptor becomes non-responsive. While to date inversion has not been proposed, it would be important to see whether inversion is important under these conditions to cause desensitization of the chemokine receptor. In other words is inversion responsible for desensitization. There is much data to suggest that a very different mechanism (β-arrestin) is responsible and this needs to be addressed. As this work may be very important for chemotaxis field.

LPS can stimulate integrin activation leading to a stop signal for immune cells mediated through TLR4. While the authors argue that there is chemokine receptor inversion, there may be a modifying role for integrins and other effectors (Ca levels etc). As such the LPS experiments becomes very hard to interpret.

There is a growing view that chemokines function in a hierarchical model with some being turned off while others become activated. It would be good to consider this potential reason for inversion.

While this is very exciting cell biology, I fail to understand how this contributes to a bigger picture of the immune response. The authors in the first paragraph of discussion suggest that this is critical for a good immune response, however an alternative interpretation could be that LPS subverts the immune response and does not allow immune cells to come to the nidus of infection. Without experiments to see whether the inversion is better or worse for infection outcome, the discussion is purely speculative.

Additional data files and statistical comments:

The experiments are performed with a high level of sophistication.

---

## [Author Response]

Responses to the evaluation of the revised submission:

We reiterate that there is potential to test your hypothesis further for chemokine receptors CXCR1 (SLLGN) and CXCR2 (SLLGN), which have a first helix lacking the GXXXN motif, and which are more similar b2 adrenergic receptor (IVFGN), which did not display the signatures of inversion. It would be very interesting to learn if these receptors would continue to function after strong LPS signaling, suggesting that cells expressing these receptors (including neutrophils) might still respond to endogenous chemokines in the context of bacterial sepsis.

It was not clear to us from the last round of review that determining whether CXCR1 and CXCR2 are subjected to RAT was a major concern of Reviewer #1. We would like to emphasize that unlike posttranslational modifications of proteins such as phosphorylation, topological regulation of a transmembrane protein cannot be determined by a single assay. It took us more than two years to gather all the evidences supporting RAT of CCR5. It will take at least as long to determine whether two other proteins are subject to RAT. While we appreciate and share the curiosity of the reviewer, the requested studies for sure cannot be accomplished within the timeframe of revision of the manuscript.

Reviewer #2:1) In response to my original point (3), the authors have simply removed all data regarding the G44L and N48L mutants because they could not be used to support their model. This substantially weakens the paper in two ways (and calls into question their interpretation of analogous mutants analyzed in their earlier paper). First, the notion that this motif is a key determinant of RAT simply cannot be supported because there are now no data provided in the manuscript to probe this. Second, they now have no experiments that actually perturb RAT by manipulating translocation to convince a reader that the effects can indeed be traced to altered protein topogenesis.

We disagree with the reviewer on this point. In addition to altering the direction through which the nascent polypeptide of a transmembrane protein is translocated across membranes during its synthesis, mutations made in the GXXXN motif could also affect folding of the protein after its synthesis is completed. This appears to be the case for CCR5. Unlike wild type CCR5 with the inverted topology produced in cells treated with ceramide or LPS that reached plasma membranes, the mutant CCR5 in which the GXXXN motif was disrupted was retained in the ER. As pointed out by the reviewer during the last round of review, the ER retention of the mutant proteins made it hard to interpret the results of some assays used to measure topology of CCR5. While we are disappointed that we may not be able to use the mutant protein as a tool to study RAT of CCR5, this limitation does not prevent us from drawing the conclusion that ceramide inverts the topology of wild type CCR5.

Unlike CCR5, we were able to demonstrate the critical importance of the GXXXN motif in ceramide-induced RAT of TM4SF20, because the G22L and N26L mutations disrupting the GXXXN motif only affected translocation but not folding and functions of the protein after its synthesis. This conclusion is supported by our observations that these mutants adopt the inverted topology regardless of ceramide treatment, and is active in stimulating cleavage of CREB3L1, the same function performed by wild type TM4SF20 with the inverted topology produced in cells treated with ceramide. We are thus curious how the reviewer reached the conclusion that the current study “calls into question their interpretation of analogous mutants analyzed in their earlier paper”.

2) The authors' response to my original point (4) is not consistent with a rather large body of data on how protein translocation works. I had pointed out that if their SNAP-CCR5 construct contains a signal peptide at the N-terminus, the signal would enforce translocation of the N-terminus into the lumen thereby ensuring the canonical CCR5 topology. The authors countered that the downstream TMD "prevails over...the signal peptide". This might be feasible if the sequence between the signal and TMD was relatively short and unstructured; however, the SNAP tag is neither short (being ~185 amino acids long) nor unstructured. Because topogenesis occur co-translationally (very well established), the signal peptide will initiate translocation long before the TMD is even synthesized (also very well established). By the time the TMD emerges from the ribosome, the entire SNAP tag (~20 kD) will be in the lumen. It is exceedingly unlikely that the TMD can 'override' this situation and somehow cause the lumenal SNAP tag to be retrieved to the cytosol. If this is indeed the claim, the available data is simply too weak to overturn (or ignore) the considerable literature on this issue. For this reason, I feel the authors are almost certainly wrong in their interpretation. Because they nonetheless see RAT by their assay, I would strongly suspect a flaw in their assay rather than ignoring well-established principles of how signal peptides and protein translocation work.

We agree with the reviewer that our data cannot be explained by the established model on protein translocation. However, we have to point out that the current model is established through in vitro translation of model substrates the translocation of which is constitutive and dependent on Sec61. While this model is the important first step toward understanding protein translocation, it by no means explains translocation process of all transmembrane proteins. Neither does this model explain how translocation of certain transmembrane proteins is inverted by ceramide. Along this line, we would like to point out a recent discovery showing that the first transmembrane helix of certain GPCRs is translocated through the EMC complex but not Sec61. These new discoveries suggest that there are other unknown mechanisms governing protein translocation. We would like to assure the reviewer that delineating the underlying translocation mechanism of RAT is a top priority in our lab. We hope that our new studies may expand the current model on protein translocation to address the concern of the reviewer.

3) The authors' response to point (6) actually argues against them. I had pointed out that immunogold EM labeling has a spatial precision of at best 20-30 nm, which would pose a challenge for determining whether an epitope was on one or the other side of a 5 nm thick membrane. I also noted that some of their gold clusters were ~200 nm away from the plasma membrane, making me think they were observing intracellular antigens in these cases. The authors countered that they felt an antibody sandwich could indeed reach 200 nm away (how is unclear given the known dimensions of an antibody). Regardless, if one does believe the precision is 200 nm, it is an even worse technique for determining topology where 5 nm resolution is needed. The authors cite earlier studies using immunogold EM for topology, but previous ill-advised uses of a method is not a strong argument for using it again now.

We still do not understand this criticism of the reviewer. After fixation, proteins should be fixed into their position so there should be no movement of the protein across plasma membranes during EM analysis. We thus do not understand why a 5-nm resolution is required to determine whether the epitope is extracellular or cytosolic. According to this critic, one could never determine whether the nucleus of a cell is intracellular or extracellular through EM, as its distance toward plasma membranes is much greater than 200 nm.

Minor Comments:1) The fluorescence micrographs remain of poor resolution and one cannot judge in most cases whether the staining actually represents surface localization as claimed (e.g., Figure 1 and Figure 2–figure supplement 1).

For immunofluorescent microscopy performed in the absence of cell permeablization, the fluorescent signal can only be detected if the epitope fused at the C-terminus of CCR5 is extracellular. When the same experiment is performed in permeabilized cells, in addition to the cell surface staining, intracellular staining of the epitope-tagged CCR5 should also be present because of the overexpression.

2) The authors claim that gold clusters are sometimes almost 200 nm away from the surface on the extracellular side as well. However, in each instance shown in the paper, these clusters seem to have membrane patches under them.

We are not sure the hazy signals surrounding the gold cluster are membrane patches.

Reviewer #3:The authors have added some sentences to the discussion and modified Figure 6 (summary) but in the end really did not address the questions I asked.The manuscript suggests that chemokine receptors could be regulated by inversion. If that were the case than as a cell is migrating at some point more inversion would occur to allow it to stop at its destination. Adding LPS to the system simply complicates the model and may represent a hijacking of the inversion by bacterial products. It would be nice to know whether inversion ever blocks a chemotaxing cell without having to invoke LPS. In figure 5B there is some suggestion this may be the case as fewer cells are migrating when FB1 is added to chemotaxing cells (about 35% inhibition).My point is that the paper as it stands is not really about chemotaxis but about how LPS might block chemotaxis. It’s really the pathology of a system for which the physiology has not been worked out. Does inversion play a role under normal chemotaxis to stop cells? I think this would be important for this paper.Additional data files and statistical comments:My comment is the same as in my first review. What is the physiologic importance of inversion of a chemokine receptor?

In this study, we never suggest that topological inversion of CCR5 is responsible for its chemokine ligand such as CCL5 to inactivate CCR5. This is because only LPS but not CCL5 was demonstrated to induce synthesis of dihydroceramide in macrophages. We are pleased that the reviewer agreed that our study demonstrated how LPS might block chemotaxis. However, we do not understand why the reviewer considered only chemokine but not LPS-induced inactivation of CCR5 as physiologically important, as LPS-induced inhibition of macrophage chemotaxis contributes to development of sepsis.

As pointed out in Discussion of the manuscript, we realized that CCR5 can be desensitized by its ligand through arrestin-mediated internalization. Thus, even if we demonstrate that CCL5 inverts the topology of CCR5, we have to address which pathway contributes more to inactivate CCR5. To answer this question, we have to make a mutant CCR5 resisting dihydroceramide-induced RAT by locking the topology into CCR5(A). Unfortunately, so far we are unable to find such a mutant. Thus, understanding the molecular details of the regulatory mechanism behind RAT may be required before reagents can be developed to test this hypothesis. Obviously, the study suggested by the reviewer cannot be accomplished within the timeframe of revision of the manuscript.

Responses to the decision letter after peer review:

Reviewer #1:[…] The authors have raised the issue by drawing in their interpretive drawing that this mechanism also allows phagocytes to locate microbes, but based on the sequence information, this doesn't seem to be the case and the LPS treated phagocytes will also be blinded to microbes after about 16 hours. This would give this mechanism a broader impact on leukocyte chemotaxis than the previously defined hierarchies. This would seem to have implications for challenges with clearing extracellular microbes in sepsis.

We would like to thank the reviewer for his positive comments. We agree with the reviewer that our model shown in Figure 6 of the original manuscript could be misleading. We mean to show that after contacting LPS, macrophages stop migrating further toward chemokines so they can combat bacterial infection at the current location through mechanism such as secreting proinflammatory cytokines but not necessarily through enhanced phagocytosis of bacteria. In the revised manuscript, we redraw the figure to remove any implication of phagocytosis of bacteria by macrophages. We also clarify that point, and emphasize that RAT of the chemokine receptors may be responsible for impaired clearance of bacterial infection during sepsis when we discuss the figure in the first paragraph of the Discussion section.

Reviewer #2:This manuscript arrives at the remarkable conclusion that CCR5 (and certain other GPCRs) can be made in a fully inverted topology in response to altered ceramide levels. The functional consequence of this inversion is the reduced biological activity of the receptor, which in the case of CCR5 involves macrophage migration toward its ligand, CCL5. Due to the striking nature of these claims, we read the manuscript with great interest. While there is no doubt that ceramide affects CCR5 in some manner, numerous conceptual and experimental flaws in the study's design and interpretation greatly weaken the argument that the basis of these effects is topology inversion. When the below points are considered together, it would seem that much of the data argue for ceramide affecting CCR5 trafficking, not topology. I would therefore strongly encourage the authors to reconsider publication of this study until the data for an effect on topogenesis is examined more carefully and thoroughly. Indeed, it is noteworthy (and rather surprising) that not a single experiment in the paper actually looks at CCR5 topogenesis at the ER, opting instead for relatively blunt end-point assays prone to alternative interpretations as outlined below.Major concerns:1) The assays in which the C-terminus of CCR5 is tagged and monitored by cell impermeable labelling are only sufficient to qualitatively demonstrate the topological location of the tagged terminus, but cannot accurately report on CCR5 topogenesis as a whole. It is easily conceivable that drastic changes in lipid composition within the cell result in misfolding or incomplete insertion of some TMDs such that the C-terminus winds up in the ER lumen at some low level. Given that these are all performed with an overexpressed protein, it is also conceivable that a misfolded population would "escape" the quality control machinery and successfully reach the PM. For these reasons, it is important to carefully quantify the amount of surface-exposed 'inverted' CCR5 relative to total cellular CCR5. One way to do this would be to use biotin-SNAP to label the surface exposed inverted CCR5, then use avidin pulldowns and blots to determine what proportion of total cellular CCR5 was surface labeled. Controls using pre-labeling with fluorescent SNAP label prior to biotin labeling could verify that labeling efficiencies are close to quantitative. Such an experiment would also verify that what is being detected is actually full length CCR5, not a fragment.

An experiment very similar to that proposed by the reviewer was shown in Figure 2H of the original manuscript (Figure 2G of the revised manuscript). The result shows that majority of full length CCR5(B) produced in ceramide-treated cells can be labeled by a thiol-reactive cell surface biotin labeling reaction thereby precipitable by streptavidin beads. These results support the hypothesis that CCR5 produced in ceramide-treated cells is also localized on plasma membranes.

2) In several of the critical experiments aimed at demonstrating topology inversion, the authors use glycosylation as an assay for topology. While N-linked glycosylation is a reasonable proxy for topology, O-linked glycosyation is problematic. The reason is that this reaction occurs in the Golgi on the folded protein. Thus, the absence of O-linked glycosylation could be due to a failure of the protein to reach the Golgi and/or altered folding, and not necessarily indicative of topology. Furthermore, I did not see where the authors provided evidence that O-linked glycosylation is not impacted by 16 h of ceramide treatment. This would seem to be a critical control for any assay relying on an endogenous biochemical reaction.

The data shown in Figure 2H of the original manuscript (Figure 2G of the revised manuscript) clearly demonstrated that the unglycosylated CCR5 produced in ceramide-treated cells reached cell surface. This observation is inconsistent with the interpretation that the lack of the glycosylation is caused by retention of the protein in the ER.

Unlike N-linked glycosylation, the carbohydrate attachment to proteins through O-linked glycosylation is different among individual protein. Even for the same protein, the O-linked glycosylation reaction could be different when the protein is expressed in a different cell. The extreme heterogeneity of this reaction makes it impossible to perform a control experiment to determine the effect of ceramide on O-linked glycosylation in general.

3) G44L and N48L are used as mutants intended to confine topogenesis to the inverted form. However, these mutants are problematic for three reasons. First, inspection of the crystal structure of CCR5 reveals that they both face the interior of the folded protein, with N48 interacting with D76, and the absence of a side chain on G44 being important for avoiding clashes with L80. Thus, mutating either residue to Leucine will very likely disrupt the folding of the protein, and hence might cause it to be primarily retained in the ER. This would readily explain why it is not O-glycosylated by Golgi enzymes (Figure 2B) and why none of it is labelled at the surface in Figure 2E without having to invoke an inverted topology. Second, the non-conservative mutations that substantially increase hydrophobicity is ill-advised as it may have effects on topology independently of the GXXXN motif. Instead, mutating this motif to residues of comparable hydrophobicity is preferable to more cleanly illustrate that N is crucial to ceramine-induced effects. Even this may well be confounded by effects on protein structure/stability. Third, the increased hydrophobicity of the G44L or N48L mutations is actually predicted to favour the correct orientation with the N-terminus facing the extracellular side (see PMID 9151664), not the inverted orientation as proposed by the authors. For all of these reasons, it is quite unlikely that mutants are acting in the way the authors propose.

Unlike wild type CCR5, we did not perform experiments to carefully examine the localization of CCR5(N48L) in the original manuscript. Following the analogy with TM4SF20, we assumed that the mutant protein is locked into the inversed topology, as the apparent molecular weight of the mutant protein is identical to unglycosylated CCR5(B) regardless of ceramide treatment. After reading the comments of the reviewer, we performed a cell surface biotin labeling experiment similar to that shown in Figure 2G to determine the localization of the mutant protein. Unlike the results obtained from wild type CCR5, we were surprised to find out that the mutant protein barely reached cell surface regardless of ceramide treatment. This observation and results from immunofluorescent microscopy suggest that CCR5(N48L) is primarily localized in the ER. Thus, the reviewer is correct in that the data showing the lack of the O-linked glycosylation and the absence of the N-terminal SNAP labeling of the mutant protein may not be used to support our hypothesis. As a result, we remove all data using the mutant CCR5 from the original manuscript, and discuss our new finding regarding the mutant CCR5 in paragraph six of the Discussion in the revised manuscript. We thank the reviewer for pointing out this important problem in our original manuscript. However, we would like to emphasize that removing these data does not alter our conclusion regarding ceramide-induced topological inversion of wild type CCR5, because both CCR5(A) produced in the absence of ceramide and CCR5(B) generated in ceramide-treated cells reach the cell surface.

4) The SNAP-CCR5 construct used in Figure 2E and Figure 2—figure supplement 2 would seem to directly disprove the authors' conclusions. As stated in the main text, the construct contains a SNAP tag (~20 kD) at the N-terminus, and according to the Materials and methods, this was obtained from CisBio. The sequence of the construct from CisBio indicates that the construct contains the N-terminal signal sequence from human CD8. This would explain how the SNAP tag domain actually gets translocated across the ER membrane. Thus, the topology of this construct is determined by the N-terminal signal sequence, not the first transmembrane domain as in the untagged construct. Rather remarkably, the SNAP-CCR5 also shows strong ceramide-induced exposure at the cell surface (Figure 2E) and loss of O-linked glycosylation (supplement). Since this construct's topology is not determined by the first transmembrane domain, it would seem that both the surface exposure and glycosylation are affected by ceramide for other reasons. The simplest interpretation is retention of the protein in the ER after ceramide treatment, which would also explain why the N48L is constitutively not only the surface and why it is constitutively not glycosylated. The authors seems to have completely overlooked the fact that this construct has a signal sequence. If they have removed it, then it is not stated in the Materials and methods (and it would be very hard to explain how the N-terminus could be translocated across the ER membrane).

The reviewer assumes that addition of an N-terminal signal peptide will override the RAT signal generated by the GXXXN motif present in the first transmembrane helix. We actually had the same hypothesis when we studied RAT of TM4SF20. Astonishingly, the translocation signal generated by the GXXXN motif in the first transmembrane domain prevails over that produced by the signal peptide, as fusing a prolactin signal peptide at the N-terminus of TM4SF20 did not affect ceramide-induced RAT of TM4SF20 (Chen et al., 2016, cited in the manuscript). Thus, we are not surprised at the finding that the presence of a signal peptide at the N-terminus of SNAP-CCR5 does not affect RAT of the fusion protein. This is one of the reasons why we are unable to make a mutant CCR5 or TM4SF20 that is locked in the A form, resisting ceramide-induced topological inversion. We make this point clearer when we present the data shown in Figure 2D in paragraph five of the Results section of the revised manuscript. This point is also discussed in paragraph three of the Discussion section of the revised manuscript.

5) The cysteine mutations intended to be a reporter of topology are problematic because several cysteines on the cytosolic side of CCR5 are palmitoylated. Thus, mutating these may well have consequences for trafficking of the receptor (see PMID 21819967 for a review), confounding any interpretations from this approach.

The result shown in Figure 2G of the original manuscript (Figure 2F of the revised manuscript) clearly demonstrates that CCR5(A) with all cysteines at the cytosolic side mutated still reached cell surface in the absence of ceramide treatment. This observation should address the concern of the reviewer.

6) The authors use immunogold labelling to infer topology of CCR5 in macrophages (Figure 3J-3L). This is conceptually problematic and the observations do not seem to fit with their favoured interpretation. Given that their primary antibody recognizes an epitope close to the membrane and the secondary antibody appears to be a gold-conjugated Fab (not entirely clear in the Materials and methods), the label can be up to ~21 nm away from the epitope (~14 nm for an IgG, and ~7 nm for the Fab). This distance increases to ~28-30 nm if the secondary antibody is a whole IgG. In either case, the label can easily be on either side of the membrane regardless of the topology because the membrane thickness is only ~5 nm. Hence, without extensive numbers and statistical power, one simply cannot get the 5 nm resolution needed to infer topology using a labelling method with a precision of 20-30 nm.

Did the reviewer suggest that immunogold EM is not suitable to determine whether an epitope is at the intracellular or extracellular side of plasma membranes? Using immunogold EM, a previous study (Singer et al., 2001, cited in the manuscript) clearly showed that the N-terminal domain of CCR5 is exclusively localized at the extracellular side of plasma membranes in cells under resting state. They did not observe any intracellular staining. This technique has been frequently used to determine the topology of transmembrane proteins localized on plasma membranes (For example, PMID: 26797119).

Remarkably however, they do see a difference. But inspection of the micrograph in Figure 3K shows that the gold particles are ~200 nm or more away from the plasma membrane. As should be obvious from the above considerations, this cannot be compatible with labelling of a cell surface protein. Many of the examples in the supplement are also too far away (although a scale bar was not provided). Thus, the simplest explanation is that after LPS treatment, the CCR5 that is labelled is in intracellular compartments, perhaps the ER (which is often close to the plasma membrane). However, the membrane morphology is not retained very well in their EM images, so one cannot really evaluate this.

The CCR5-specific gold cluster signal is caused by binding of multiple gold-conjugated secondary antibodies to anti-CCR5. As a result, the distance between these signals and plasma membranes are longer than that calculated by the reviewer. This pattern of CCR5 staining was also observed in an earlier study (Singer et al., 2001, cited in the manuscript). Consistent with this explanation, we observed that the distance between extracellular CCR5 signal and plasma membranes in resting cells is not too much different from that between intracellular CCR5 signal and plasma membranes in LPS-treated cells.

We also added the scale bar in images shown in Figure 3—figure supplement 3 of the revised manuscript. The scale bar was somehow lost when we converted our figure into PDF format during last submission.

A second major issue is that they only observe an average of ~1.3 labels per cell. It is difficult to know what to make of this given how much signal is seen in the fluorescent images using the same antibody. For these reasons, the authors' interpretation of the immuno-EM cannot be supported from the data they present.

As stated in the manuscript, we observed both gold cluster and single gold particle-labeled signal in the EM images. Only the gold cluster signal is considered as CCR5-specific as such signal was never observed in CCR5^-/-^ macrophages. Some CCR5 was labeled by a single gold particle, as their number in wild type macrophages was higher than that in CCR5^-/-^ macrophages. However, the specificity of such labeling was difficult to determine, as these particles did exist in CCR5^-/-^ macrophages. As a result, we only counted the gold cluster signal. This number should be smaller than that of CCR5 molecules, as it did not include CCR5 labeled by a single gold particle. We make this point clearer when we discuss these results in paragraph ten of the Results section of the revised manuscript.

7) In the absence of the immuno-EM experiment, the macrophage data is very weak in establishing any role for topology inversion of CCR5 in LPS-stimulated macrophage migration. Figure 3A to 3I simply establishes that LPS reduces surface levels of CCR5 and does not speak at all to its topology. The same is true for Figure 4. All of these data are equally consistent with LPS causing intracellular retention (or endocytosis) of CCR5 in a dihydroceramide dependent manner. Thus, the effects on migration seen in Figure 5 simply reflects the fact that there is less CCR5 on the surface of these macrophages.

As stated in our comments above, we believe that our immuno-EM data do provide a critical piece of evidence to support out hypothesis. In addition, our immunofluorescent microscopy results shown in Figures 3D and E suggest that CCR5 was localized on plasma membranes regardless of LPS treatment. Thus, our data about endogenous CCR5 expressed in macrophages are consistent with our model that LPS-induced dihydroceramide production triggers RAT of CCR5.

Minor Comments:1) The fluorescence microscopy is poor resolution and often over-exposed (at least in the images I was provided) so it is very hard to judge localization (surface versus intracellular locations).

The immunofluorescent signals generated in non-permeabilized cells were much weaker than that produced in permeabilized cells. Since the exposure was kept at the same level for all images, we had to overexpose the images of the permeabilized cells in order to see a clear signal of non-permeabilized cells. In the revised manuscript, we include immunofluorescent microscopy results of the permeabilized cells with normal intensity in Figure 3—figure supplement 2. These results show that CCR5 in macrophages is primarily localized on plasma membranes regardless of LPS treatment.

2) Introduction – the authors state that "it remains unclear how GPCRs can adopt such a membrane topology without the signal peptide." This is not accurate, as the study of how an N-terminal transmembrane domain directs topology has been extensively studied. Such sequences are termed signal anchors because they act as both a signal sequence and transmembrane anchor. The features that determine their topology has been investigated by extensive mutagenesis many years ago (see PMID 15461443 for a review, and PMIDs 9151664, 8557050, and 1985975 for some of the primary papers), and quite a bit has also been done on the molecular basis of their insertion (see PMID 10943843).

We rewrite this part of Introduction in the first paragraph of the revised manuscript to incorporate this information.

3) Throughout the paper, the authors essentially assume that an inverted GPCR would traffic to the cell surface. This seems very unlikely because the extracellular domain facing the cytosol would not form disulphide bonds and therefore not fold correctly, while the C-terminus facing the outside would not be palmitoylated. A more nuanced discussion is merited.

We do not ASSUME that GPCR with the inverted topology reaches cell surface. This conclusion is directly supported by the data shown in Figure 1 and Figure 2G of the revised manuscript. However, we agree with the reviewer that a better discussion is needed to clarify this point. We now show in paragraph four of the Discussion section that our data regarding inaccessibility of the antibodies against extracellular regions of CCR5(A) with the GPCR configuration in ceramide-treated cells may be interpreted by ER retention of the protein. This model may also be consistent with the observation that ceramide treatment blocked O-linked glycosylation of CCR5, as this post-translational modification takes place in the Golgi complex. However, this model cannot explain our observations that ceramide treatment exposed C-terminus of CCR5 to the extracellular space. Nor is this model consistent with the observation that significant amount of CCR5 in cells treated with ceramide or macrophages treated with LPS, which stimulated production of dihydroceramide, was localized on cell surface. Thus, taking all of our data together, we believe that topological inversion through RAT is the better interpretation of our results.

4) The surface labelling kit cited in the Materials and methods labels amines, not sulfhydryls. Some clarification is needed here about how the experiment was actually done.

We thank the reviewer for pointing out this important mistake we made in the original manuscript. We actually labeled cell surface cysteine residues with 0.25 mg/ml EZ-link maleimide-PEG2-biotin (Thermo Fisher Scientific) and then immunoprecipitated the labeled proteins with the Pierce Cell Surface Labeling Kit. We correct this mistake in the section of Materials and methods.

Reviewer #3:This study proposes a very nice mechanism of chemokine receptor inversion to explain why cells may stop chemotaxing. While interesting, the biological situation chosen is a difficult one to understand i.e., LPS stimulation of peritoneal macrophages prevents migration to CCL5. I will ask a few questions that would need to be answered with additional experiments.There is much work to demonstrate that when high enough concentrations of a chemokine for sufficient time leads to downregulation of the receptor or the receptor becomes non-responsive. While to date inversion has not been proposed, it would be important to see whether inversion is important under these conditions to cause desensitization of the chemokine receptor. In other words is inversion responsible for desensitization. There is much data to suggest that a very different mechanism (β-arrestin) is responsible and this needs to be addressed. As this work may be very important for chemotaxis field.LPS can stimulate integrin activation leading to a stop signal for immune cells mediated through TLR4. While the authors argue that there is chemokine receptor inversion, there may be a modifying role for integrins and other effectors (Ca levels etc). As such the LPS experiments becomes very hard to interpret.There is a growing view that chemokines function in a hierarchical model with some being turned off while others become activated. It would be good to consider this potential reason for inversion.

The current study suggests that LPS-induced topological inversion of CCR5 may be one of the mechanisms for LPS to inhibit chemotaxis mediated by the receptor. This conclusion is supported by our observations that FB1, which blocked LPS-induced topological inversion of CCR5 by inhibiting synthesis of dihydroceramide, restored the chemotaxis reaction of macrophages exposed to LPS. However, we agree with the reviewer that the evidence we provided is not sufficient to demonstrate that topological inversion of CCR5 is the primary mechanism for LPS to desensitize CCR5, as dihydroceramide may activate the arrestin and/or integrin pathway mentioned by the reviewer, or any unknown pathways that inactivate CCR5. The only way to address the question of the reviewer is to make a mutant CCR5 resisting dihydroceramide-induced RAT by locking its topology into the GPCR configuration (CCR5(A)). Unfortunately, fixing topology of CCR5 and TM4SF20, another protein subjected to RAT, into their A form appears to be challenging. Just adding a signal peptide at the N-terminus of these proteins does not block ceramide/dihydroceramide-induced topological inversion. Thus, understanding the molecular details of the regulatory mechanism behind RAT may be required before reagents can be developed to address the question raised by the reviewer. This point is now discussed in paragraph three of the Discussion section. We would like to reassure the reviewer that we are vigorously pursuing study to address this question, but owing to the lack of the key reagent, we are unable to include these studies into the current manuscript.

While this is very exciting cell biology, I fail to understand how this contributes to a bigger picture of the immune response. The authors in the first paragraph of discussion suggest that this is critical for a good immune response, however an alternative interpretation could be that LPS subverts the immune response and does not allow immune cells to come to the nidus of infection. Without experiments to see whether the inversion is better or worse for infection outcome, the discussion is purely speculative.

We agree with the reviewer that our model shown in Figure 6 is speculative at this time. We have revised our discussion in the first paragraph of the Discussion in the revised manuscript to demonstrate this point. As suggested by the reviewer, we also discuss in this paragraph the potential contribution of this mechanism to inhibit macrophage migration during sepsis.